# Cilia locally synthesize proteins to sustain their ultrastructure and functions

Kai Hao 🄳 [1,2], Yawen Chen[1,2], Xiumin Yan 🄳 [3,5 ✉] & Xueliang Zhu 🄳 [1,2,4,5 ✉]

Cilia are microtubule-based hair-like organelles propelling locomotion and extracellular liquid flow or sensing environmental stimuli. As cilia are diffusion barrier-gated subcellular compartments, their protein components are thought to come from the cell body through intraflagellar transport or diffusion. Here we show that cilia locally synthesize proteins to maintain their structure and functions. Multicilia of mouse ependymal cells are abundant in ribosomal proteins, translation initiation factors, and RNA, including 18 S rRNA and tubulin mRNA. The cilia actively generate nascent peptides, including those of tubulin. mRNA-binding protein Fmrp localizes in ciliary central lumen and appears to function in mRNA delivery into the cilia. Its depletion by RNAi impairs ciliary local translation and induces multicilia degeneration. Expression of exogenous Fmrp, but not an isoform tethered to mitochondria, rescues the degeneration defects. Therefore, local translation defects in cilia might contribute to the pathology of ciliopathies and other diseases such as Fragile X syndrome.

[1] State Key Laboratory of Cell Biology, Shanghai Institute of Biochemistry and Cell Biology, Center for Excellence in Molecular Cell Science, Chinese Academy of Sciences, 320 Yueyang Road, 200031 Shanghai, China. [2] University of Chinese Academy of Sciences, 100049 Beijing, China. [3] Ministry of Education-Shanghai Key Laboratory of Children's Environmental Health, Institute of Early Life Health, Xinhua Hospital, Shanghai Jiao Tong University School of Medicine, 200092 Shanghai, China. [4] School of Life Science, Hangzhou Institute for Advanced Study, University of Chinese Academy of Sciences, 310024 Hangzhou, China. [5] These authors jointly supervised this work: Xiumin Yan, Xueliang Zhu. ✉email: yanxiumin@xinhuamed.com.cn; xlzhu@sibcb.ac.cn

Proteins are traditionally known to be translated in the cytosol or on the endoplasmic reticulum and then reach their functional sites through intracellular trafficking or by diffusion[1]. Nevertheless, it is recently recognized that mRNAs can be delivered to polarized subcellular compartments such as dendrites and axons in the form of messenger ribonucleoprotein (mRNP) granules to locally translate proteins. Such a mechanism enables the control of proteomes at spatiotemporal precision in response to regional or micro-environmental cues[2,3]. Currently, local translation has been recognized to fine tune a variety of biological activities including embryonic development, mitosis, cell migration, and neuronal plasticity[2,4].

Cilia or flagella are polarized organelles protruding out of the cell surface. The ciliary membrane covers an axoneme of nine microtubule (MT) doublets emanated from a specialized centriole, or the basal body, that is anchored to the plasma membrane[5,6]. Immotile cilia serve as sensors to environmental signals such as light, force, odorants, and hedgehog molecules, whereas motile cilia or flagella function as propellers for locomotion of protozoa, sperms, and many invertebrates or extracellular fluid flow above animal tissue surfaces through rhythmical rapid beating[5,7]. Defects in ciliary functions result in genetic disorders termed ciliopathies. Primary ciliary dyskinesia (PCD), for instance, manifests as a syndromic disease including situs inversus, infertility, hydrocephalus, recurrent sinusitis, and chronic bronchitis[8,9].

Cilia are subcellular compartments gated from cytoplasmic contents by a diffusion barrier at the transition zone (TZ)[10]. Therefore, ciliary proteins are generally believed to be synthesized primarily in the cytoplasm. They then translocate in the form of complexes or alone into the ciliary shaft, mainly through a train-like motor-driven intraflagellar transport (IFT) machinery and rarely, usually for proteins less than 50 kDa, by diffusion[11–13]. These translocation mechanisms, however, might not fully account for the supply of ciliary proteomes, especially those of motile cilia, which require hundreds of proteins to construct their elegant and intricate architecture[6,14]. In fact, only a small pool of the ciliary structural proteins, including α/β-tubulin, has been shown to move along anterograde IFT trains[15,16]. Even for tubulin, the known docking sites on IFT trains appear to be insufficient for the rapid flagellar regeneration of *Chlamydomonas reinhardtii*[15–17]. Notably, ribosomal proteins are readily identified in proteomes of motile cilia purified from *C. reinhardtii*[18,19], human bronchial epithelial cells[14], and mouse ependymal cells (mEPCs)[20]. We thus reasoned that local protein synthesis might be a supplemental source of ciliary proteins.

In this study, we demonstrated that multicilia of cultured mEPCs contain protein translation machinery and actively synthesize proteins such as tubulin. Ciliary mRNP granules contained Fmrp, an RNA-binding protein (RBP) critical for neuronal plasticity by regulating local translation in neurites[2,21]. Furthermore, depriving the ciliary pool of Fmrp resulted in degeneration of ependymal multicilia.

## Results

**Ependymal motile cilia contain translation machinery.** Protein synthesis requires the translation machinery composed of the ribosome and various eukaryotic initiation factors (eIFs)[22]. To confirm that ribosomal proteins can be readily identified from motile cilia, we purified multicilia from cultured mEPCs (Fig. 1a)[23], performed shotgun mass spectrometry[24], and hit 95% of known ribosomal components (Fig. 1b). Furthermore, the ciliary proteome also covered 66% of the components of eIFs (Fig. 1b), implying the existence of translation initiation machinery[22]. Immunoblotting confirmed that ribosomal protein

Rpl4, eIF3 components Eif3d and Eif3h were enriched by >7.5 fold in multiciliary preparations over the abundant cytosolic protein Gapdh (Fig. 1c). In another sets of independent immunoblotting results, in which we loaded equal amount of total proteins each lane, Rpl4, Eif3f, and Eif3h were enriched by >6 fold over Tom20 (Supplementary Fig. 1a), a mitochondrial protein[25].

We then performed immunofluorescent (IF) staining to directly detect ciliary localization of ribosomal components. Rpl10a, Rpl11, and Rps3, ribosomal proteins critical for the mRNA docking channel[26], were detected as puncta in ependymal multicilia (Fig. 1d). To exclude the possibility of nonspecific staining of the antibodies, we expressed GFP-tagged Rpl11 and Rps3 in mEPCs and observed their punctate distributions along the ciliary shaft as well (Fig. 1e).

Next we examined whether multicilia contained RNA. Acridine orange (AO) is a nucleic acid-specific fluorescent dye that distinguishes RNA (red fluorescence) from DNA (green fluorescence)[27]. To our excitement, we readily detected strong red AO fluorescence that was sensitive to RNase A treatment in multicilia, whereas the green fluorescence observed in the nucleus was insensitive to RNase A (Fig. 1f, g and Supplementary Fig. 1b). Similarly, Pyronin Y (PY), a nucleic acid-specific dye preferably recognizing RNA (Supplementary Fig. 1c)[28], also produced prominent RNase-sensitive fluorescent signals in multicilia (Supplementary Fig. 1d, e). These results suggest that ependymal multicilia are abundant in RNA.

We reasoned that, if the ribosomal components resided in ependymal cilia (Fig. 1b–e) in the form of ribosomes, we would be able to detect ribosomal RNA (rRNA) as well. We performed both single-molecule fluorescent in situ hybridization (smFISH)[29] and conventional FISH[30] and indeed detected RNase-sensitive fluorescent signals of 18 S rRNA in ependymal multicilia (Fig. 1h and Supplementary Fig. 1f). Quantification indicated that the RNase A treatment reduced the relative ciliary intensity of 18 S rRNA by 62 fold on average in the smFISH experiments (Fig. 1h, i).

Of eIFs identified by our mass spectrometry (Fig. 1a, b), we confirmed punctate ciliary localization of multiple eIF3 subunits, including core subunits Eif3f, Eif3h, Eif3m, and a peripheral subunit Eif3b (Fig. 2a-c)[22]. Moreover, the mRNA 5′ cap-binding protein Eif4e and the key scaffold protein Eif4g[22] were also detected in ependymal cilia (Fig. 2a, d). Super-resolution microscopy indicated that Eif3m and Eif4g were distributed as puncta close to axonemal outer MT doublets (Fig. 2e). Taken together, we conclude that ependymal multicilia contain ribosomes, eIFs, and RNA.

**Ependymal multicilia locally synthesize proteins.** To verify local translation occurred in ependymal cilia, we performed metabolic labeling of Puromycin (Puro), a structural analog of aminoacyl-tRNAs that is incorporated into the C termini of nascent polypeptide chains to allow antibody detection (Fig. 3a)[31,32]. To minimize possible diffusion of puromycylated peptides[33], we performed both a 5 min and a 10 min pulse labeling and observed clear ciliary IF signals of Puro (Fig. 3b and Supplementary Fig. 2a). The IF signals were punctate and uneven along the ciliary shaft, thus unlikely attributed to puromycylated peptides diffused into the cilia from the cell bodies. Quantification revealed that the mean relative intensity exceeded that of the mock-treated cilia by >8.3 fold, whereas the average intensity was reduced by >2.8 fold in the presence of protein synthesis inhibitor Cycloheximide (CHX)[34] (Fig. 3b, c and Supplementary Fig. 2a, b), indicating a tight correlation of the Puro IF signals with protein translation.

Cytoplasmic puromycylated peptides have been shown to diffuse rapidly into the nucleus[35]. Although the barrier function of TZ[10] would probably make ciliary compartment less accessible

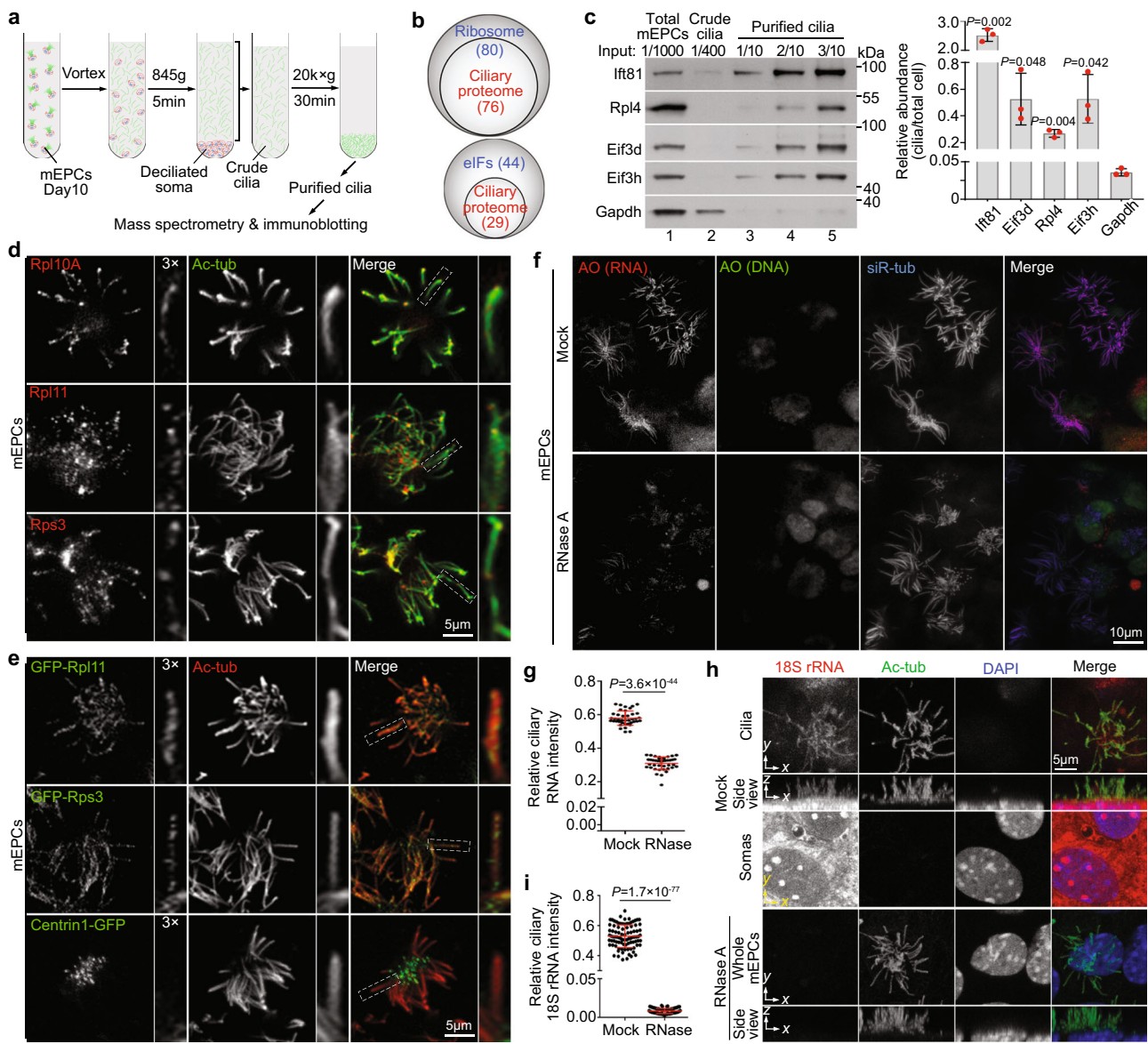

**Fig. 1 Ependymal multicilia contain ribosomal components and RNA.** Pooled results are presented as mean ± s.d. Two-tailed student's *t*-test.
**a** Experimental scheme for cilia purification. **b** Venn diagrams showing that ciliary proteome contained most ribosomal and eIF components. Proteomics of purified multicilia identified 76 out of 80 known ribosomal proteins and 29 out of 44 eIF components as compared with AmiGo 2 open-source. **c** Rpl4, Eif3d, and Eif3h were enriched in purified multicilia. IFT81 and Gapdh served as ciliary and cytosolic markers, respectively. Band intensities of lane 4 relative to lane 1 were measured ($n = 3$ biologically independent experiments). Refer to Supplementary Fig. 1a for an additional set of results. **d** Endogenous Rpl10A, Rpl11, and Rps3 localized in multicilia of cultured mEPCs. Typical cilia were magnified to show details. Acetylated tubulin (Ac-tub) labeled ciliary axonemes. **e** GFP-tagged Rpl11 and Rps3 displayed ciliary localization. mEPCs were infected with lentivirus at 1 day before serum starvation (day −1) to express GFP-Rpl11, -Rps3, or Centrin1. **f, g** Multicilia were abundant in RNA. Cilia are marked with siR-tub and the cells were stained with acridine orange (AO) after mock or RNase A treatment. Representative z-projected images from cilia-containing z-stacks are shown (**f**). Refer to Supplementary Figure 1b for z-projected images for cell bodies of the same samples. Ciliary AO fluorescent intensity was normalized to that of siR-tub (**g**). 40 multiciliated mEPCs were quantified in each condition. **h, i** Ependymal cilia contained 18 S rRNA. mEPCs fixed at day 10 were treated with or without RNase A, followed by smFISH to detect 18 S rRNA. Ac-tub and DAPI, respectively, labeled cilia and nuclei. For the representative mock-treated sample, z-projected images from z-stacks above and below the dashed yellow line, respectively, are presented to show fluorescent signals in cilia and soma (**h**). For the representative RNase-treated sample, the z-projected images were from the entire z-stacks (**h**). Ciliary FISH fluorescent intensity was normalized to that of Ac-tub ($n = 3$ independent experiments). 30 multiciliated mEPCs were measured in each experiment and condition.

to cytoplasmic puromycylated peptides, the possibility of peptide diffusion still needed to be excluded. To further confirm that puromycylated nascent peptides were indeed locally synthesized in cilia, we initially isolated multicilia-containing apical cortexes of mEPCs by adapting a previously reported method for oviduct multicilia (Fig. 3d). Consistently[36], multicilia of the isolated cortexes retained basal bodies and still existed in bundles (Fig. 3e).

They were positive for components of the translation machinery as well (Fig. 3e). More importantly, we performed Puro-incorporation assays and still detected punctate IF signals in the multicilia (Fig. 3f). The IF signals were abolished either in the presence of CHX or in the absence of Puro (Fig. 3f).

We noticed, however, that strong IF signals of Puro emerged at the basal body area of isolated cortexes (Fig. 3f), raising a

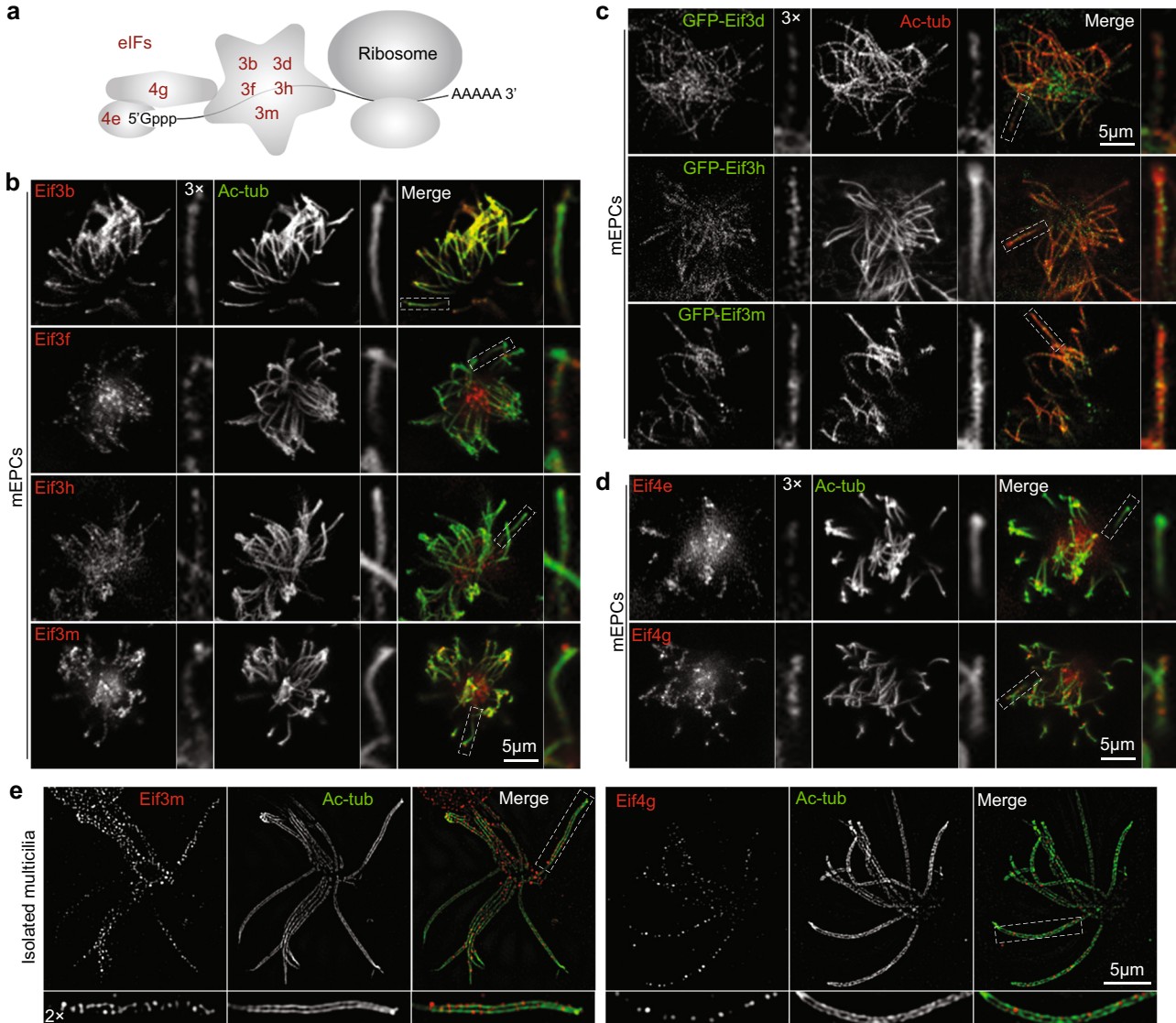

**Fig. 2 Ependymal multicilia contain eIF components. a** Diagrams illustrating locations of investigated eIFs. **b** Ciliary localization of endogenous Eif3 components in day-10 mEPCs. Typical cilia were magnified to show details. Ac-tub labeled axonemes. **c** Ciliary localization of GFP-tagged Eif3 components. Cultured mEPCs were infected with lentiviral particles at day −1 to express GFP-fusion proteins and fixed at day 10 for immunostaining. **d** Ciliary localizations of endogenous Eif4E and Eif4G. **e** Ciliary localization of Eif3m and Eif4g. Isolated multicilia were imaged under three-dimensional structured illumination microscopy (3D-SIM).

possibility that some of these IF signals might had diffused into the cilia. To obtain solid evidence for the ciliary in situ protein synthesis, we explored whether isolated ciliary shafts without the basal body could display active protein translation. We did not use the protocol illustrated in Fig. 1a because vortexing suspended mEPCs would break the ciliary shafts and cause the leakage, even loss, of ciliary contents important for protein synthesis. Instead, we added the $Ca^{2+}$-containing deciliation buffer[23] directly to intact mEPCs in culture flasks and used the shearing force of a shaker to detach TZ-containing ciliary shafts from their cell bodies (Fig. 3g). Immunostaining confirmed that the majority (76.5%) of the isolated ciliary shafts contained TZ and lacked the basal body, judged by the presence of Cep290, a protein located at the bottom region of TZ[37], and the absence of basal body marker Odf2[38] at the ciliary base (Supplementary Fig. 2c, d). 23.2% of them were negative for both markers, whereas only 0.3% of them were positive for both Cep290 and Odf2 (Supplementary Fig. 2c, d). When the isolated ciliary shafts were subjected to Puro labeling from 5 to 30 min, they readily displayed clear time- and

translation-dependent IF signals of Puro (Fig. 3h,i), indicating that they still actively synthesize proteins. The mean relative intensities of ciliary Puro IF signals exceeded those of CHX-treated samples by 4.4 fold at 5 min, 4.0 fold at 10 min, and 6.2 fold at 30 min (Fig. 3i). These results verify the existence of active local translation in ependymal cilia and also suggest that the cilia have a high protein synthesis capacity.

**FMRP forms puncta in the central lumen of ependymal multicilia.** As protein synthesis requires mRNA, we speculated that certain mRNP granules might help to deliver mRNAs from the cytoplasm into ependymal multicilia, analogous to other cases of local translation[3,39]. We compared our ependymal cilia proteome (Fig. 1a) with an RBP database from RBPDB open-source for candidate RBPs of ciliary mRNP granules and identified 45 overlapping RBPs (Fig. 4a). To exclude proteins that were not or only weakly expressed in multiciliated cells, we examined our previous cDNA microarray results of differentiating mouse

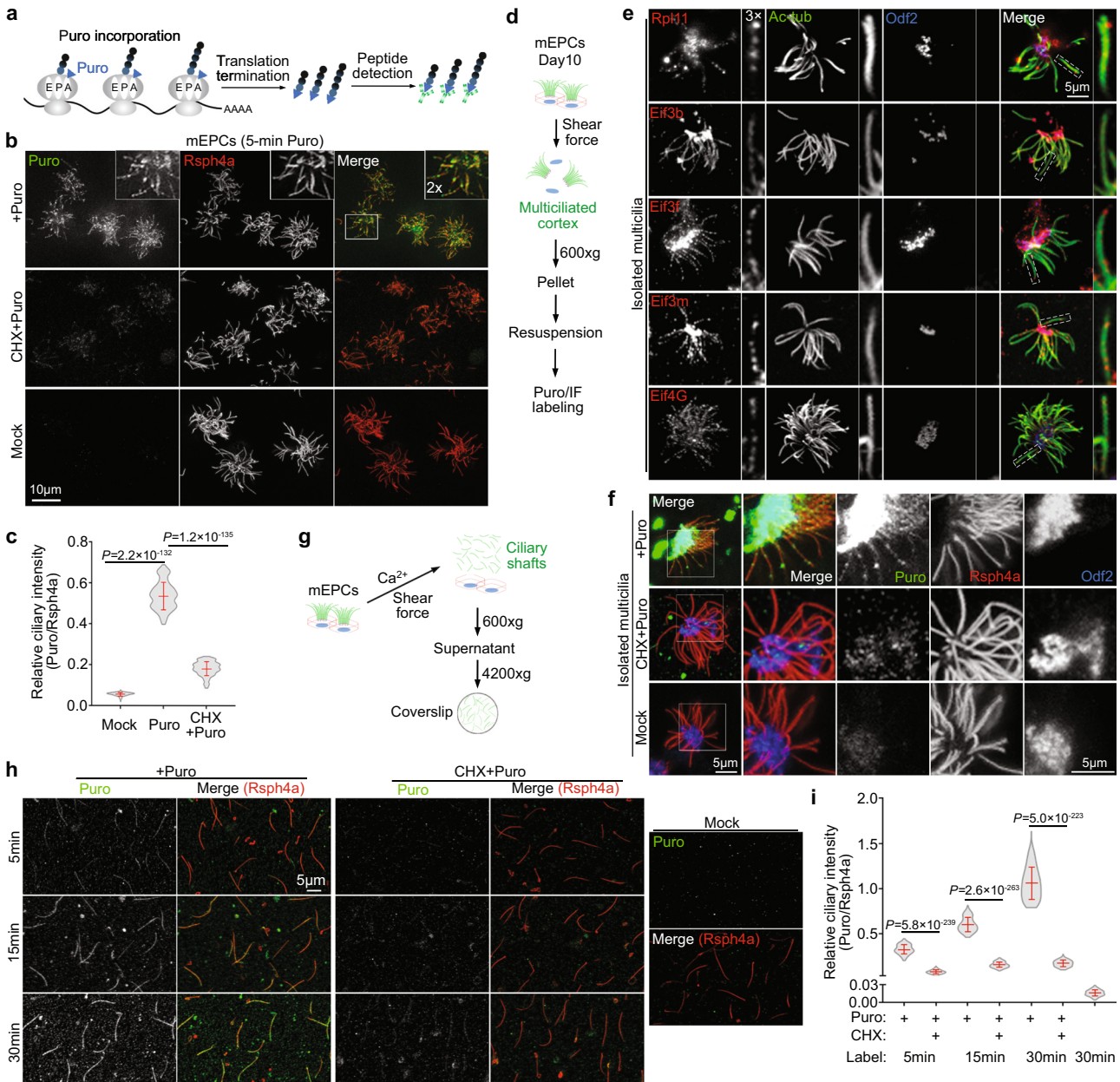

**Fig. 3 Ependymal cilia display local protein synthesis.** Relative intensities of ciliary Puro were from three independent experiments. The bars and errors represent mean ± s.d. Two-tailed student's *t*-test. **a** Diagram illustrating metabolic labeling of nascent peptides with Puromycin (Puro). Puromycylated peptides were visualized with an anti-Puromycin antibody. The E-, P-, and A-sites in the ribosome are depicted. **b, c** Protein translation-dependent Puro IF signals were readily detected in multicilia. Day-10 mEPCs were pulse-labeled with Puro for 5 min with or without Cycloheximide (CHX) and then processed for confocal imaging (**b**). Rsph4a, a radial spoke subunit, served as multiciliary marker[71]. The framed region was magnified to show details. In **c**, 150 multiciliated mEPCs were quantified in each condition. Ciliary intensity of Puro was normalized to that of Rsph4a. **d** Diagram illustrating the isolation of multicilia-containing apical cortexes. **e** Multicilia of the isolated cortexes retained translation machinery components and basal bodies. Ac-tub and Odf2 served as ciliary and basal body markers, respectively. **f** Protein translation-dependent Puro IF signals were detected in isolated multicilia. Pulse labeling was performed as in **b**. **g** Diagram illustrating the isolation of ciliary shafts. Multiciliated mEPCs were cultured to day 10 in flasks. After the culture medium was replaced with $Ca^{2+}$-containing deciliation buffer[23], the flasks were shaken horizontally to release individual ciliary shafts. The ciliary shafts in the supernatant were then spun onto poly-L-lysine-coated coverslips for further analyses. **h, i** Isolated ciliary shafts displayed local protein translation. Isolated ciliary shafts attached to poly-L-lysine-coated coverslips were pulse-labeled with Puro for 5, 15, and 30 min with or without CHX. Mock-treated samples served as negative control. After fixation, the samples were immunostained for Rsph4a and subjected to confocal imaging (**h**). In **i**, 300 cilia were quantified in each condition. For each ciliary shaft, the total ciliary IF intensity of Puro was normalized to that of Rsph4a.

tracheal epithelial cells (mTECs)[40] and found that they contained mRNA hybridization signals of only 10 candidate RBP genes, among which 6 candidate genes displayed upregulation and persistent expression during the multiciliation (Fig. 4b). Interestingly, three of them, Fmrp, Caprin1 (cell cycle-associated protein 1), and RNA helicase Upf1, are involved in local mRNA localization[21,41], whereas the remaining three, Pspc1, Nucleolin, and U1-70K, are nuclear RBPs located in paraspeckles, nucleoli, and spliceosomes, respectively[42–44]. As Fmrp is the best documented mRNP component critical for mRNA delivery into

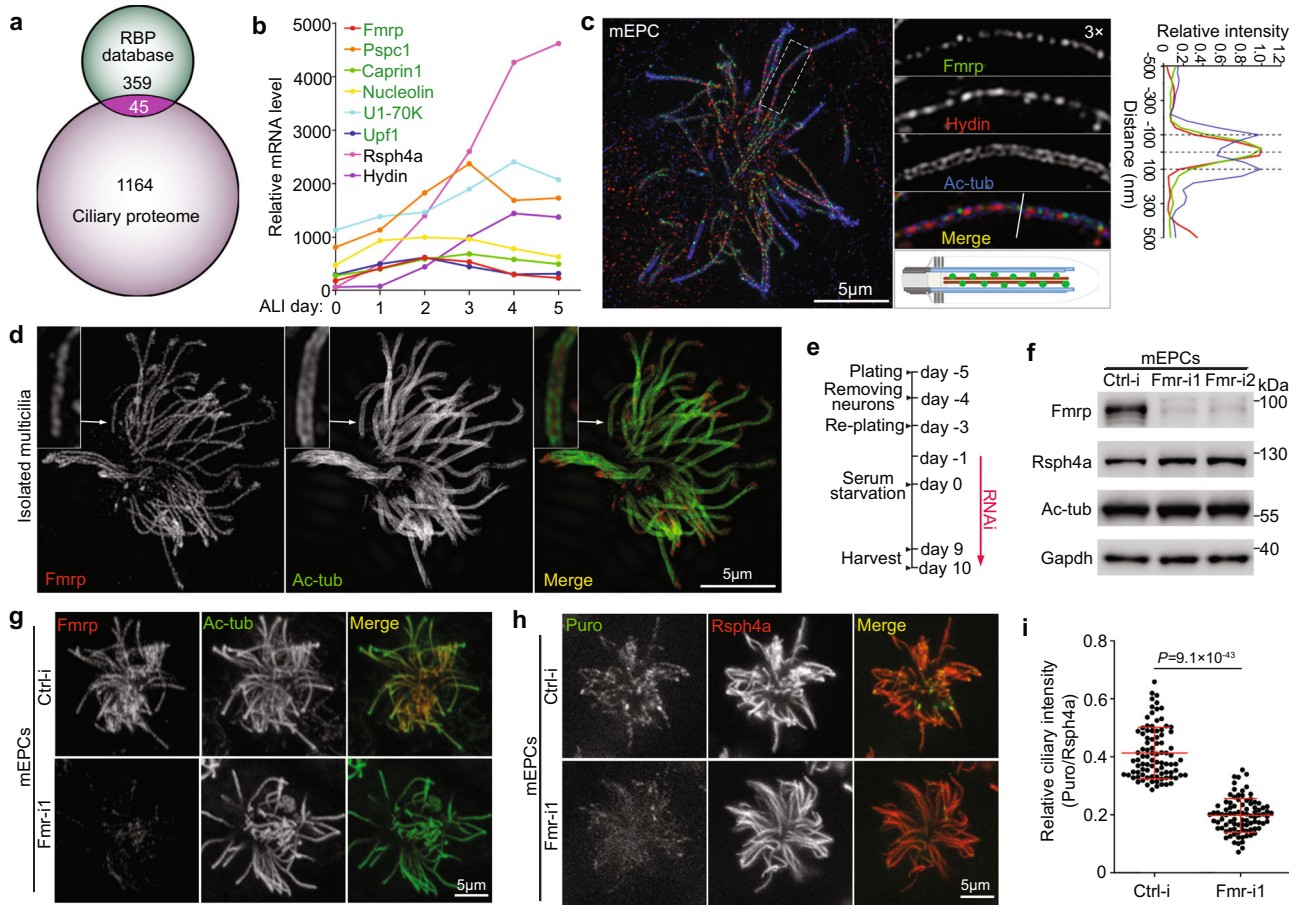

**Fig. 4 Fmrp is enriched in the central lumen and important for local translation of ependymal cilia. a** Venn Diagram showing overlapping proteins between our cilia proteome and an RBP database. Protein hits with ≥5 unique peptides in the proteome were used for comparison. **b** Gene expression patterns of the indicated RBPs (green) during multiciliation. Data were from our previous cDNA array results of mTECs[40]. Rsph4a and Hydin were used to show expression patterns of typical motile cilia-related genes. **c** Fmrp was enriched in the ciliary central lumen as puncta. Day-10 mEPCs were imaged with 3D-SIM. Ac-tub and Hydin labelled axonemes and central pairs, respectively. A typical cilium was magnified to show details. Line scan at the indicated region of the magnified inset shows colocalization of Fmrp with Hydin. The diagram depicts the central lumen localization of Fmrp. **d** Representative 3D-SIM images of isolated day-10 ependymal multicilia showing central lumen distribution of punctate Fmrp. **e** Experimental scheme for RNAi in mEPCs. mEPCs were transfected with a control siRNA (Ctrl-i) and two siRNAs against *Fmr1* (Fmr-i1, -i2) and collected at day 10 for immunoblotting (**f**) and immunostaining (**g**) or day 9 for Puro labeling (**h**). As multicilia formation occurs from after day 3, this protocol is expected to deplete Fmrp prior to multiciliation. **f** Efficient depletion of Fmrp. Gapdh served as loading control. The levels of Rsph4a and Ac-tub were used to reflect efficiencies of multiciliated cell differentiation and multicilia formation, respectively. **g** Confocal images showing depletion of Fmrp markedly reduced the ciliary IF signals of Fmrp. **h, i** Depletion of Fmrp impaired ciliary local protein translation. mEPCs transfected with Ctrl-i or Fmr-i1 were subjected to Puro labeling (see Fig. 3a) at day 9. Rsph4a served as ciliary marker in confocal imaging (**h**). Quantification results (**i**) were pooled from three independent experiments. 30 multiciliated mEPCs were measured in each experiment and condition. For each cell, the total ciliary IF intensity of Puro was normalized to that of Rsph4a. The bars and errors represent mean ± s.d. Two-tailed student's *t*-test.

dendrites and axons of neurons for local translation[2,21] and also interacts with Caprin1[45], we chose to focus on it in the following assays. Interestingly, when we used super-resolution imaging to assess detailed localization, Fmrp was found to distribute along the central axis of axonemes as discrete puncta (Fig. 4c). Its distribution pattern resembled Hydin (Fig. 4c), a protein associated with the central pair of MTs[23]. In isolated multicilia (Fig. 3d), Fmrp also nicely distributed in the central region of axonemes (Fig. 4d). Therefore, Fmrp is enriched in the axonemal central lumen.

**Fmrp is critical for ciliary local translation probably by delivering mRNAs.** As Fmrp also functions as a translational repressor[46,47], we clarified its ciliary functions. We reasoned that, if the ciliary Fmrp resided in mRNP granules responsible for mRNA delivery into the cilia, its depletion would result in

reduced ciliary local translation. On the other hand, if it functioned to directly regulate, i.e., repress, ciliary local translation, its depletion would result in increased ciliary local translation. We thus efficiently depleted FMRP by RNAi through two independent siRNAs prior to multicilia formation (Fig. 4e, f). The depletion of Fmrp did not reduce the band intensities of Rsph4a and Ac-tub (Fig. 4f), suggesting that Fmrp is dispensable for both multiciliated cell differentiation and multiciliogenesis. Consistently, multicilia normal in morphology were common in the Fmrp-depleted mEPCs at day 10 (Fig. 4g). When Puro labeling (Fig. 3a) was performed; however, ciliary IF signals of Puro became markedly declined in the Fmrp-depleted cells (Fig. 4h, i). The average ciliary IF intensity dropped by 2.1 fold as compared to the control mEPCs (Fig. 4i). These results thus further strengthen the idea that Fmrp is involved in the formation and ciliary transportation of mRNP granules for the ciliary local translation.

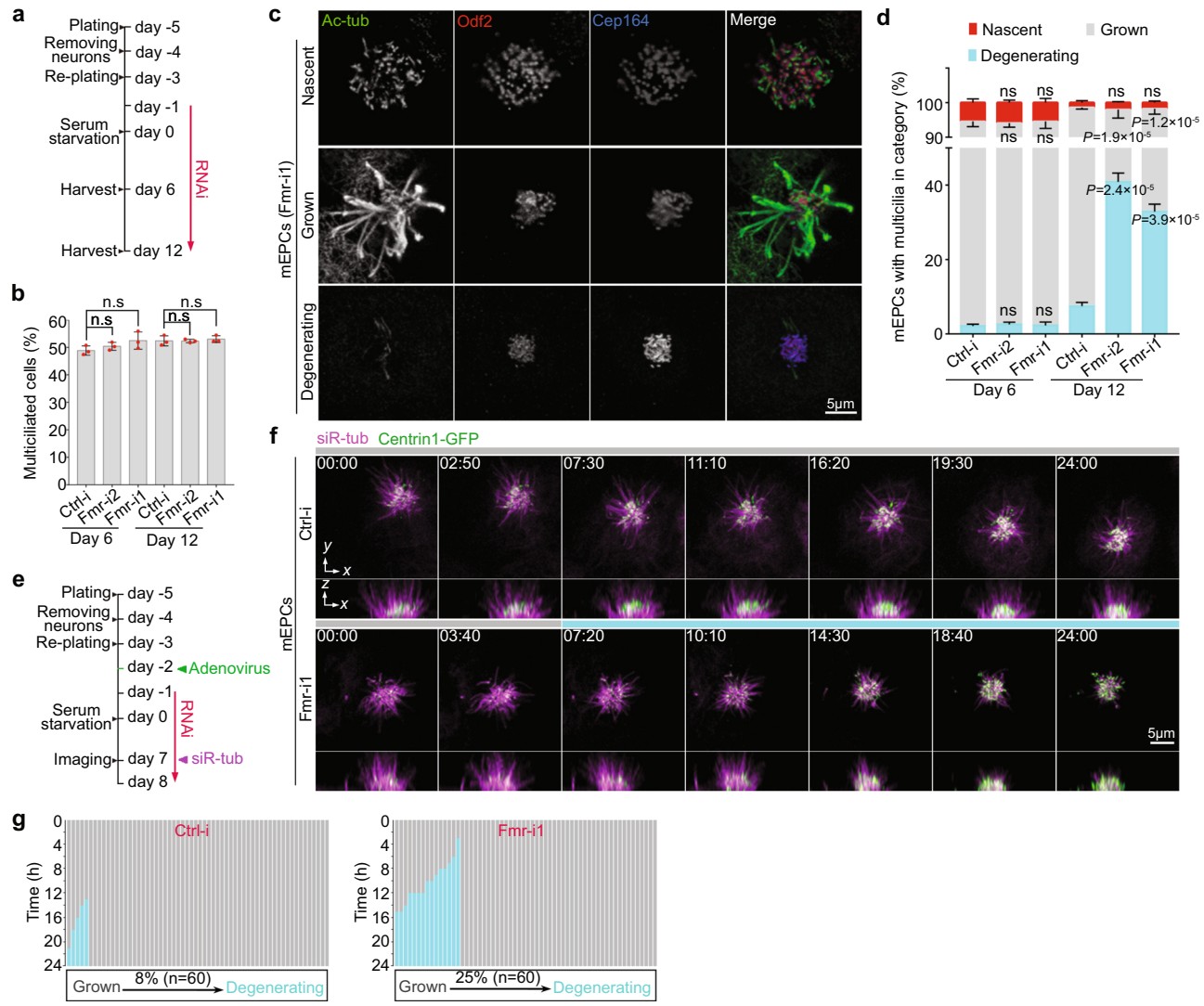

**Fig. 5 Depletion of FMRP promotes degeneration of ependymal multicilia.** Pooled data were from three independent experiments and are presented as mean ± s.d. Two-tailed student's *t*-test against control populations: ns, no significance. **a** Experimental scheme. mEPCs transfected with siRNA (Ctrl-i or Fmr-i1) were harvested at day 6 or 12 to assess the status of multiciliation. **b** Fmrp was dispensable for multiciliated cell formation. 1280 (Ctrl-i, Day 6), 1456 (Fmr-i2, Day 6), 1501 (Fmr-i1, Day 6), 1350 (Ctrl-i, Day 12), 1225 (Fmr-i2, Day 12), and 1360 (Fmr-i1, Day 12) cells from three biologically independent experiments were scored. **c** Representative patterns of multicilia in day-12 mEPCs transfected with Fmr-i1. Odf2 and Cep164 served as basal body markers. Nascent: short multicilia with scattered basal bodies; Grown: long multicilia with packed basal bodies; Degenerating: a few uneven multicilia with overwhelming numbers of tightly packed basal bodies. **d** Depletion of Fmrp increased mEPCs with degenerating multicilia at day 12. mEPCs were scored following the criteria in **c**. 635 (Ctrl-i, Day 6), 789 (Fmr-i2, Day 6), 895 (Fmr-i1, Day 6), 883 (Ctrl-i, Day 12), 690 (Fmr-i2, Day 12), and 899 (Fmr-i1, Day 12) cells from three biologically independent experiments were scored. **e** Experimental scheme for tracing dynamic changes of multicilia through live imaging. mEPCs were infected with adenovirus to express Centrin1-GFP, followed by siRNA (Ctrl-i or Fmr-i1) transfection. Spinning disk microscopy was performed from day 7 for 24 h at 10 min intervals after the addition of siR-tub to label axonemes. At each timepoint, z-stack images were acquired at 0.5 μm intervals for a depth of 15 μm to sufficiently cover basal bodies and ciliary tips. **f** Representative frames showing multicilia degeneration in mEPCs transfected with Fmr-i1. Both z-projected images (x-y) and their orthogonal views of 3D-reconstructed images (x-z) are presented. The dashed line marks the middle plane of basal bodies. Time stamps are in the format of hours: minutes. **g** Quantification of 60 multiciliated cells in each condition. Multicilia undergoing degeneration are marked in blue.

**FMRP is critical for the maintenance of ependymal multicilia.** To assess physiological importance of the ciliary local translation, we examined effects of Fmrp depletion in detail. The depletion of Fmrp did not alter the overall percentage of multiciliated mEPCs at day 6 and 12 (Fig. 5a, b). Nascent multicilia of ependymal cells are initially scattered throughout the apical domain. Their basal bodies then cluster together so that the grown multicilia appear as a radial bundle[48]. Nevertheless, we noticed that mEPCs containing tightly packed basal bodies but short sparse multicilia

became prominent following time upon the depletion of Fmrp (Fig. 5c, d), implying that the cells undergo cilia degeneration, i.e., pathological disassembly of multicilia over time. At day 6, the majority of multiciliated mEPCs (≥92%) contained grown multicilia (Fig. 5c, d). The ratios among mEPCs with nascent, grown, and degenerating multicilia were unchanged between the control and FMRP-depleted samples (Fig. 5c, d). At day 12, however, Fmrp-depleted mEPCs with degenerating multicilia increased by >13 fold to 33% (Fmr-i1) or 41% (Fmr-i2), accompanied with a

dramatic decrease in those with grown multicilia (Fig. 5d). By contrast, in the Ctrl-i-transfected populations, mEPCs with degenerating multicilia only occupied 7% (Fig. 5d).

To confirm that Fmrp-depleted multicilia indeed tended to undergo degeneration, we expressed Centrin1-GFP through adenoviral infection to label basal bodies and performed live imaging from day 7 for 24 h to trace dynamic changes of multicilia (Fig. 5e). We observed that grown multicilia displayed progressive shortening during the imaging in 25% ($n = 60$) of the Fmr-i1-transfected mEPCs but only 8% of the control mEPCs (Fig. 5f, g and Supplementary Movie 1). Furthermore, the Fmr-i1-transfected mEPCs usually started multicilia degeneration earlier than the control cells (Fig. 5g).

Staining for nuclear DNA and the tight junction marker Zo-1 indicated that even cells with completely degenerated multicilia, i.e., containing tightly packed basal bodies but no cilia, were morphologically normal (Supplementary Fig. 3a). Therefore, the multicilia degeneration is not due to cell death.

**Multicilia require cilia-localized Fmrp for maintenance**. We next clarified whether the ciliary degeneration (Fig. 5) was due to the depletion of ciliary Fmrp. As the ciliary compartment is generally believed to exclude mitochondria (see Supplementary Figs. 1a and 4a for the lack of ciliary enrichment of Tom20), we reasoned that, if we tethered exogenous Fmrp to mitochondria to abolish its ciliary entry and examined its rescue effect on the ciliary degeneration phenotype, we would be able to discriminate the ciliary function of Fmrp from the cytoplasmic function. We thus created Fmr-i1-insensitive adenoviral constructs to express GFP-Fmrp and GFP-mtFmrp, whose C-terminus contained the transmembrane domain of monoamine oxidase capable of ectopically targeting proteins to the mitochondrial outer membrane (Fig. 6a)[49]. The expression level of GFP-mtFmrp was similar to that of endogenous Fmrp in mEPCs (Fig. 6b, c). In the cell bodies of FMRP-depleted mEPCs, GFP-mtFmrp colocalized with Tom20, a mitochondrial outer membrane receptor[25], in discrete speckles (Fig. 6d), confirming that it is tethered onto mitochondria[49]. By contrast, GFP-Fmrp was dispersed in the cytoplasm[47] and irrelevant to mitochondria (Fig. 6d). As expected, GFP-Fmrp, but not GFP-mtFmrp, displayed multicilia localization (Fig. 6e). When the GFP-positive mEPCs transfected with Fmr-i1 were quantified at day 12, GFP-Fmrp markedly reduced the incidence of multicilia degeneration as compared to Centrin1-GFP (Fig. 6f). GFP-mtFmrp, however, displayed no rescue effect (Fig. 6f), indicating that Fmrp needs to localize into cilia to prevent ciliary degeneration.

**Ependymal multicilia contain α/β-tubulin mRNA**. Next we sought to identify ciliary mRNAs whose local translations could be involved in the ciliary degeneration. We reasoned that, if Fmrp indeed functioned to deliver mRNAs into ependymal cilia, its depletion would attenuate ciliary mRNAs as in the case of neurites[2,21]. We could therefore identify candidate ciliary mRNAs by comparing RNA deep sequencing results of cilia purified, respectively, from ctrl-i- and Fmr-i1-treated mEPCs. Ciliary mRNAs were expected to be low in abundance comparing with cytoplasmic mRNAs and susceptible to degradation during purification. To compromise between the yield of cilia and the degradation risk of their mRNAs, we chose to purify multicilia (Fig. 7a, b). We analyzed the RNA deep sequencing results and, respectively, sorted out 89 and 118 mRNAs of known cilia-related proteins that were downregulated in the FMRP-depleted samples in two sets of independent experiments (Fig. 7c and Supplementary Data 1). The majority of these candidate ciliary mRNAs

(63/89 and 92/118) encoded proteins in ciliary shafts according to gene ontology annotations of the AmiGo 2 database. Thirty six candidate mRNAs overlapped, including mRNAs of multiple α- and β-tubulin isoforms, MT regulators (*Dcdc2a* and *Dcx*), tubulin polyglutamylase and glycylase (*Ttll9* and *Ttll10*), axonemal dynein subunits (*Dnah12* and *Dnali1*), subunits of axonemal dynein-docking complexes (*Drc7* and *Ttc25*), IFT subunits (*Bbs1*, *Ift27*, and *Ift52*), other proteins involved in the structure and function of motile cilia (e.g., *Armc9*, *Cep131*, *Cfap69*, *Dnajb13*, and *Tmem17*), and even Fmrp (*Fmr1*) (Fig. 7c and Supplementary Table 1). As tubulin mRNAs are among candidate Fmrp-targeting mRNAs in neurons and other cell types[47] and local translation of tubulin was also in agreement with the ciliary degeneration phenotype of Fmrp-depleted mEPCs (Figs. 5 and 6), we performed smFISH[29] for α- and β-tubulin mRNAs in mEPCs and detected RNase-sensitive FISH signals in multicilia (Fig. 7d-g), confirming the existence of α/β-tubulin mRNAs in ependymal cilia.

**Ependymal multicilia locally translate α-tubulin**. To verify that the tubulin mRNAs were indeed translated into tubulin in ependymal cilia, we performed the metabolic Puro labeling in mEPCs (Fig. 3a), followed by proximity ligation assays (PLA) to detect ciliary puromycylated tubulin peptides (Fig. 8a)[50]. When a rabbit polyclonal antibody raised against full-length α-tubulin (anti-α-tubF; 1:800) (Fig. 8b) was paired with the anti-Puro antibody, robust punctate PLA fluorescent signals along multicilia were observed (Fig. 8c and Supplementary Movie 2). Quantification indicated that the PLA signals from both antibodies exceeded those from either antibody by over 150 fold (Fig. 8c, d), confirming that only the combination of the two antibodies could efficiently produce PLA signals. As cilia are abundant in MTs, we further assessed whether the strong PLA signals from the anti-α-tubF and anti-Puro antibodies (Fig. 8c, d) were produced specifically, i.e., through the concomitant binding of anti-α-tubulin and anti-Puro antibodies to the same puromycylated α-tubulin peptides as illustrated in Fig. 8a, or nonspecifically through their separate binding to α-tubulin molecules in axonemal MTs and any puromycylated peptides. We introduced a rabbit polyclonal antibody raised against a C-terminal peptide of α-tubulin (anti-α-tubC) (Fig. 8b). When anti-α-tubC was diluted at 1:100, it generated comparable IF signals of ependymal ciliary axonemes as anti-α-tubF did (Fig. 8b). We reasoned that, if PLA signals could be produced nonspecifically through the separate antibody binding, anti-α-tubC antibody (1:100) and anti-α-tubF antibody (1:800) should generate comparable PLA signals with anti-Puro antibody. On the contrary, as puromycylated α-tubulin peptides were expected to mostly lack epitopes of anti-α-tubC antibody, anti-α-tubC antibody would only be able to generate much weaker ciliary PLA signals than anti-α-tubF did if the PLA signals indeed required the concomitant binding of anti-α-tubulin and anti-Puro antibodies to the same puromycylated α-tubulin peptides as a prerequisite. When the PLA reactions were performed parallelly, we observed that the PLA signals from anti-α-tubF and anti-Puro antibodies exceeded those from anti-α-tubC and anti-Puro antibodies by 21.4 fold (Fig. 8c, d). Therefore, we conclude that ependymal multicilia contain puromycylated α-tubulin peptides.

The PLA reactions are highly sensitive, capable of detecting a single puromycylated peptide by principle (Fig. 8a)[50]. Therefore, although we verified that cytoplasmic puromycylated peptides did not substantially diffused into ependymal cilia during the metabolic labeling (Fig. 3 and Supplementary Fig. 2), it was still necessary to clarify the source of puromycylated α-tubulin peptides that generated the ciliary PLA signals (Fig. 8c). As the

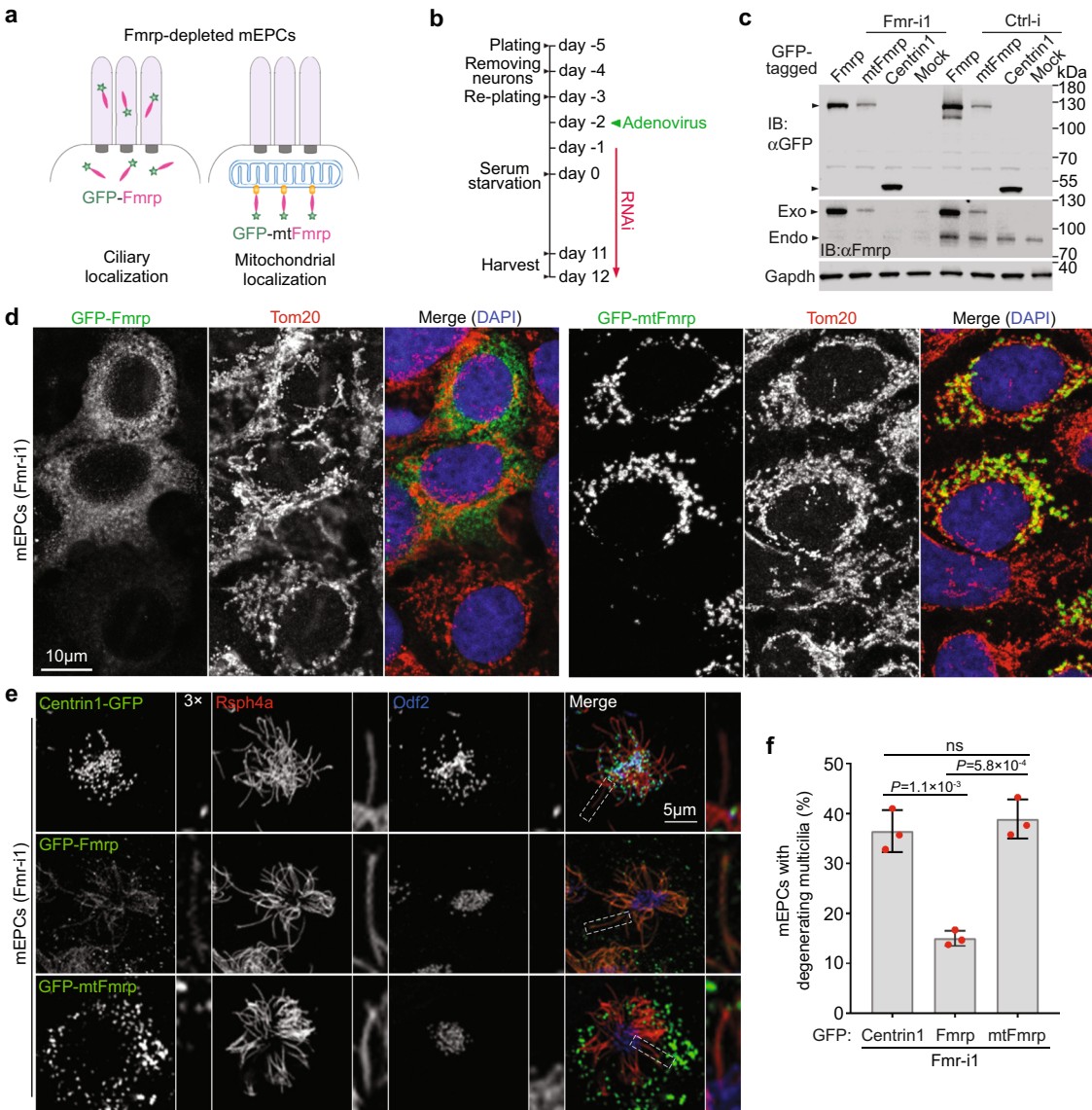

**Fig. 6 Fmrp depletion-induced multicilia degeneration is attributed to the ciliary fraction of Fmrp.** Pooled data were from three independent experiments and are presented as nean ± s.d. Two-tailed student's t-test: ns no significance. **a** Diagrams illustrating rescue experiments using adenovirus-expressed GFP-Fmrp and a mitochondrial tethered form, GFP-mtFmrp. **b** Experimental scheme of the rescue experiments. mEPCs were harvested at day 12 unless described otherwise. **c** Expression levels of GFP-tagged proteins. Gapdh served as loading control. Exo exogenous Fmrp, Endo endogenous Fmrp. **d** GFP-mtFmrp displayed mitochondrial localization in cell bodies of day-10 mEPCs. Tom20 served as mitochondrial marker. **e** GFP-Fmrp, but not GFP-mtFmrp, displayed multicilia localization in mEPCs transfected with Fmr-i1. Centrin1-GFP was used as negative control. Rsph4a and Odf2 served as ciliary and basal body markers, respectively. Typical cilia (framed) were magnified to show details. **f** Incidence of multicilia degeneration. 689 (Centrin-GFP), 762 (GFP-Fmrp), and 343 (GFP-mtFmrp) multiciliated cells from three biologically independent experiments were scored.

abundant cytosolic protein Gapdh and mitochondrial protein Tom20 were rarely detected in ependymal cilia (Supplementary Fig. 4a; also see Fig. 1c and Supplementary Fig. 1a)[20], we examined ciliary PLA signals of their puromycylated peptides by using antibodies against N-terminal Gapdh (anti-GapdhN) and full-length Tom20 (anti-Tom20F). We observed that neither anti-GapdhN antibody nor anti-Tom20F antibody produced ciliary PLA signals comparable to anti-α-tubF antibody in parallel experiments (Fig. 8e and Supplementary Fig. 4b). As intensities of PLA signals might differ among the antibodies, we converted the PLA signals to PLA spots and quantified the spot numbers as described previously[51] for unbiased statistical analysis (Fig. 8e,f and Supplementary Fig. 4b). The average number of ciliary PLA

spots from puromycylated α-tubulin peptides exceeded those from puromycylated peptides of Gapdh and Tom20 by 35.7 and 24.6 fold, respectively, whereas their average PLA spot numbers in somas were comparable (Fig. 8f). We also performed PLA assays using isolated ciliary shafts and obtained similar results (Fig. 8g, h). The average ciliary PLA spot number from puromycylated α-tubulin peptides exceeded that from puromy-cylated peptides of Gapdh or Tom20 by 29-fold (Fig. 8h). Therefore, we conclude that ependymal multicilia locally translate α-tubulin. These results also indicate that the ciliary compartment is much less accessible to cytoplasmic puromycylated peptides than compartments inside the cytoplasm such as the nucleus[35], probably due to the barrier function of TZ[10].

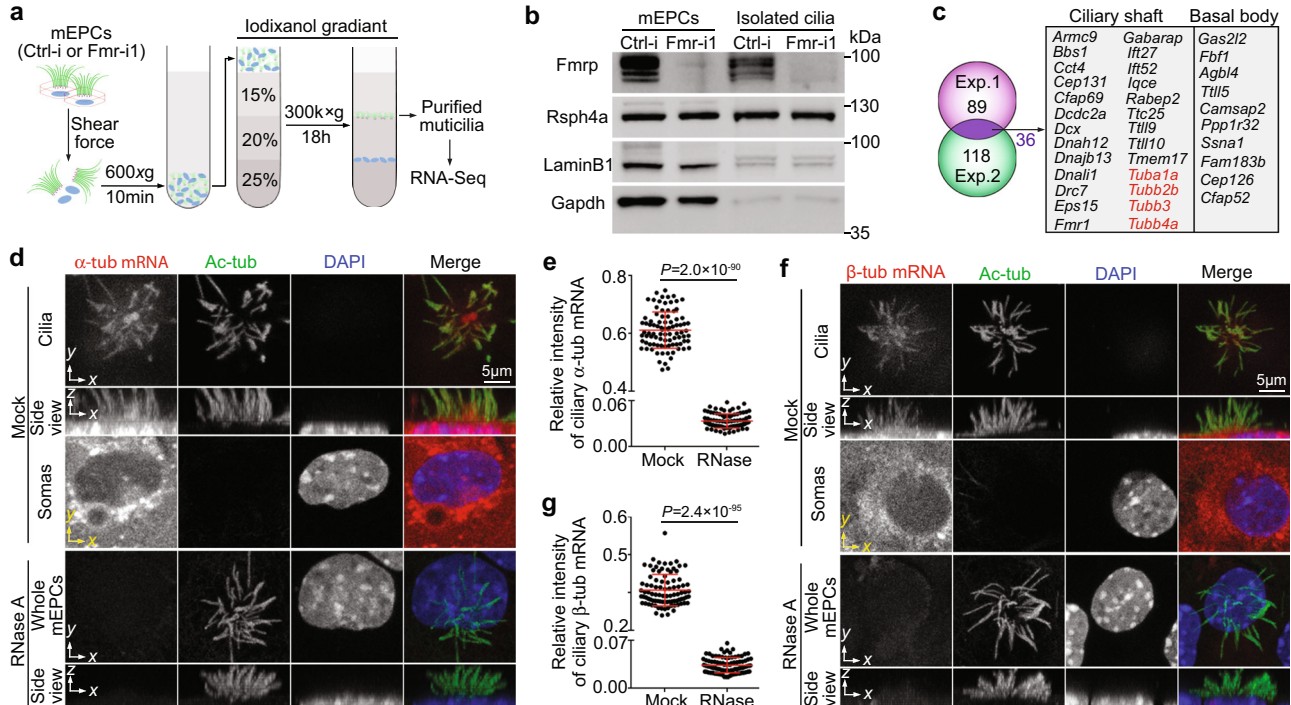

**Fig. 7 Ependymal multicilia contain α/β-tubulin mRNAs.** Pooled results are presented as mean ± s.d. Two-tailed student's *t*-test. **a** Experimental scheme for multicilia purification. Day-10 mEPCs infected, respectively, with Ctrl-i and Fmr-i1 as illustrated in Fig. 4e were subjected to shear force. The released multicilia were purified through the iodixanol gradient ultracentrifugation and used for RNA deep sequencing to identify potential ciliary mRNAs. **b** Sample quality shown by immunoblotting. For the multicilia samples, proteins collected after RNA extraction were used. Fmrp was efficiently depleted in mEPCs. Purified multicilia were largely free of cytosolic (Gapdh) and nuclear (Lamin B1) contaminants. Rsph4a served as cilia marker. **c** Candidate ciliary mRNAs. 89 and 118 mRNAs coding for cilia-related proteins were respectively found to be downregulated in the Fmr-i1 ciliary samples in two independent experiments (see Supplementary Table 1 for details). The 36 overlapping mRNAs were considered as candidate ciliary mRNAs. Tubulin mRNAs are shown in red. **d–g** Detection of ciliary α/β-tubulin mRNAs by smFISH. mEPCs fixed at day 10 were treated with or without RNase A, followed by smFISH to detect the mRNA for α-tubulin (**d**) or β-tubulin (**f**). Ac-tub and DAPI, respectively, labeled cilia and nuclei. For the representative mock-treated samples (**d, f**), z-projected images from z-stacks above and below the dashed yellow line, respectively, are presented to show fluorescent signals in cilia and soma regions. For the representative RNase-treated samples (**d, f**), the z-projected images were from the entire z-stacks. Quantification results (**e, g**) were pooled from three independent experiments. 30 multiciliated mEPCs were measured in each experiment and condition. For each cell, the total ciliary FISH intensity was normalized to that of Ac-tub.

Finally, we examined whether the ciliary local synthesis of α-tubulin was sensitive to the depletion of Fmrp. We observed that ciliary PLA signals from puromycylated α-tubulin peptides were largely reduced in FMRP-depleted mEPCs (Fig. 8i). Statistical analysis revealed a 5.1-fold reduction in ciliary PLA signal intensity as compared to Ctrl-i-treated mEPCs (Fig. 8j). Therefore, the ciliary local translation of α-tubulin requires the Fmrp-mediated mRNA delivery.

Taken together, our results suggest a mechanism in which Fmrp mediates the ciliary delivery of mRNAs through mRNP granules for ciliary local translation to facilitate the maintenance of ependymal multicilia (Fig. 8k).

## Discussion

We demonstrate that ependymal multicilia locally translate proteins, therefore revising the present dogma that proteins synthesized in the cytoplasm are the sole source of ciliary proteins. Our results suggest that most ribosomal (95%) and eIF (66%) components are distributed in the cilia (Figs. 1a-e, 2 and Supplementary Fig. 1a)[20]. The cilia were also highly abundant in RNA, including 18 S rRNA and α/β-tubulin mRNA (Figs. 1f-i, 7d-g and Supplementary Fig. 1). More importantly, we detected abundant puromycylated nascent peptides in the cilia through metabolic Puro labeling (Fig. 3 and Supplementary Fig. 2). The prominent

Puro labeling in isolated ciliary shafts (Fig. 3h, i) not only confirmed the ciliary origin of puromycylated peptides, but also indicated high activity and capacity of the ciliary local translation as well. By using PLA to detect puromycylated peptides of certain proteins, we further showed that ependymal cilia locally synthesize α-tubulin, but not non-ciliary proteins Gapdh and Tom20 (Fig. 8 and Supplementary Fig. 4). As eIF-positive puncta localized close to the axoneme (Fig. 2e), the local translation might occur at certain spatial locations. Despite these, characterization of detailed components of ciliary translational machineries, direct visualization of their localizations, and elucidation of their functioning mechanisms still require future investigations. As proteomic studies with human tracheal multicilia and green algae flagella have also identified components of the protein translational machinery[14,18], local translation appears to be an evolutionarily conserved mechanism at least in motile cilia.

What could be the physiological roles of the ciliary local translation? Firstly, our results (Figs. 5 and 6) suggest that local translation may provide replacement components for ciliary maintenance. The continuous, rapid beat of motile cilia may cause damages to their force-generating dynein motors or lesions in their intricate ultrastructure and thus require repairs. Locally synthesized components, such as tubulin (Fig. 8), might be more convenient than the IFT-imported ones especially in beating cilia. Secondly,

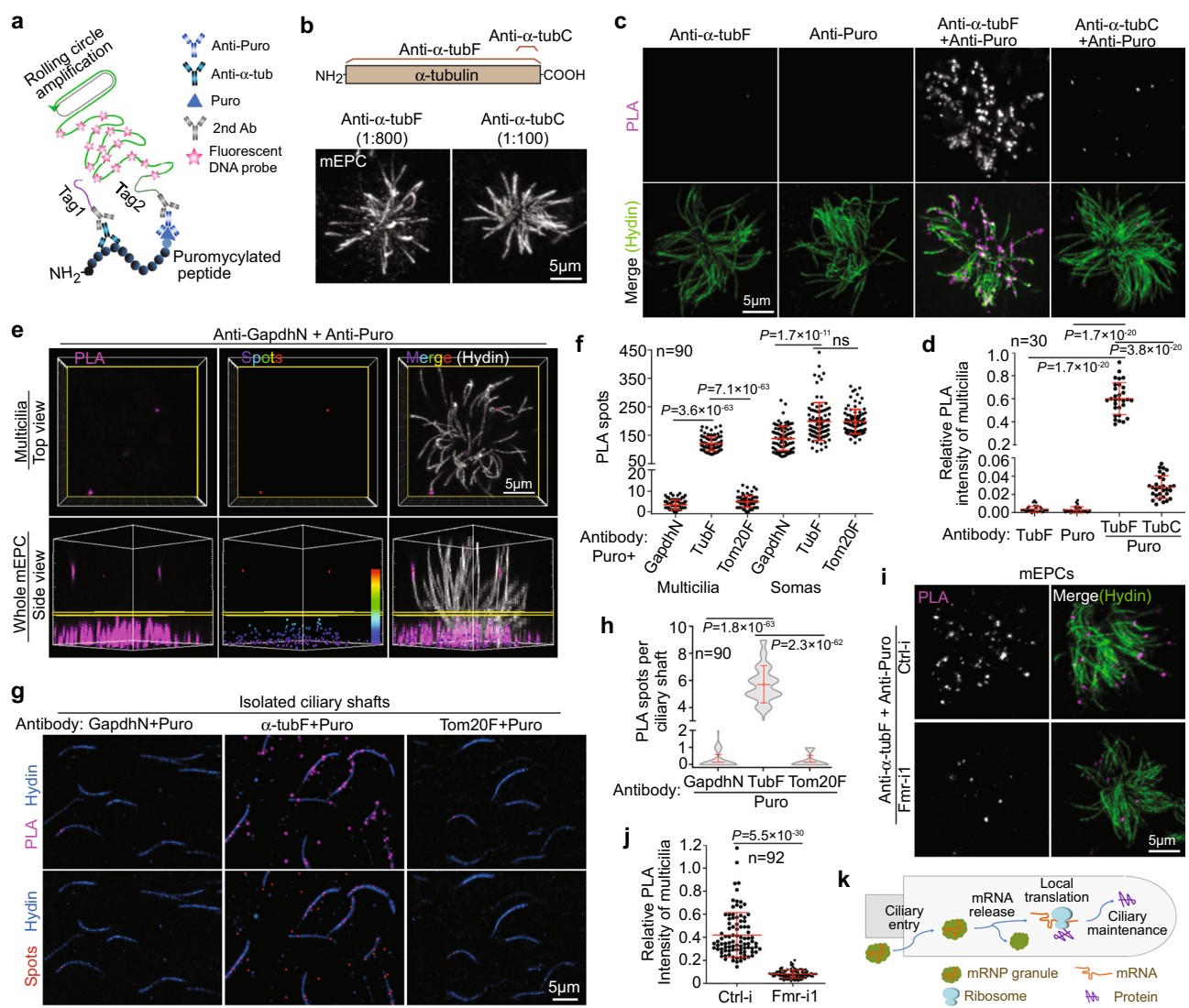

**Fig. 8 Ependymal multicilia locally translate α-tubulin.** Day 10 were used unless otherwise stated. Quantification results are presented as mean ± s.d. Those in (**f**, **h**, **j**) were pooled from three independent experiments. Two-tailed student's *t*-test: ns, no significance. **a** Diagram illustrating PLA detection of a puromycylated α-tubulin peptide. The close positioning of two oligonucleotides (tag1 and tag2) enables amplification of approximately a thousand DNA repeats that can be visualized as a bright punctum through fluorescent probe hybridization. **b** Rabbit antibodies against full-length (anti-α-tubF) and extreme C-terminal (anti-α-tubC) α-tubulin similarly decorated multiciliary axonemes at the indicated dilutions. **c**, **d** Multicilia contained puromycylated α-tubulin peptides. mEPCs were labeled with Puro for 10 min, followed by fixation and PLA using the indicated antibodies (**c**). Hydin served as ciliary marker in the representative confocal images of PLA signals in multicilia (**d**). **e**, **f** Ependymal multicilia locally synthesized α-tubulin but not Gapdh and Tom20. Representative 3D-reconstructed images showing PLA signals from anti-GapdhN and anti-Puro antibodies as a negative control (**e**). The multicilia and soma regions are divided by the yellow plane. See Supplementary Fig. 4b for images from other antibody pairs. Statistical results (**f**) were scored from software-generated color-coded PLA spots[51]. As the z-depth of soma regions varied among cells, PLA spots in somas were normalized to the value in 2000 μm[3], which was close to the average volume covered by the confocal imaging. **g**, **h** Isolated ciliary shafts locally synthesized α-tubulin but not Gapdh and Tom20. Isolated ciliary shafts were labeled with Puro for 30 min, followed by PLA (**g**, top panels). The PLA signals were converted to PLA spots (**g**, bottom panels), PLA spots were scored from well-separated ciliary shafts (**h**). **i**, **j** Depletion of Fmrp markedly reduced ciliary PLA signals of puromycylated α-tubulin peptides. mEPCs transfected as in Fig. 4e were labeled with Puro for 10 min at day 9, followed by PLA. PLA signals in representative multicilia regions (**i**) and quantification results (**j**) are shown. **k** An illustration model, Fmrp-containing mRNP granules enter ciliary shaft and release their content mRNAs for local protein synthesis to support ciliary maintenance.

comparing to the IFT-mediated protein import[11,13], local translation may serve as a local source of protein components for the construction of the complicated ciliary ultrastructure[6,14]. Thirdly, local translation may facilitate spatiotemporal remodeling of ciliary proteomes to precisely modulate ciliary functions in response to regional cues, as in the case of neurites[2,21]. Cilia are dynamic organelles even after fully grown. In *Chlamydomonas*, for instance,

20% of the flagellar proteins are exchanged within 3 h[52]. Interestingly, only some of the structural proteins, including tubulin, MT-stabilizing protein Tektin, and axonemal dynein, are prone to the turnover[52–54]. Furthermore, active protein exchanges preferentially occur at the tip and the junction of axonemal MT doublets[52,54–56]. Consistently, the candidate ciliary mRNAs identified by our deep sequencing analyses (Fig. 7c and Supplementary Table 1) include

mRNAs of MT regulators and modification enzymes (*Dcdc2a*, *Dcx*, *Ttll9*, and *Ttll10*), subunits and docking proteins of axonemal dynein arms (*Dnah12*, *Dnali1*, *Drc7*, and *Ttc25*), IFT subunits (*Bbs1*, *Ift27*, and *Ift52*), and other proteins involved in the structure and function of motile cilia (e.g., *Armc9*, *Cep131*, *Cfap69*, *Dnajb13*, and *Tmem17*). Future validation of bona fide ciliary mRNAs additional to those of tubulin and elucidation of their translational regulations and physiological importance will lead to comprehensive understandings on roles of the ciliary local translation.

We found that Fmrp localizes to ependymal cilia to support ciliary local translation and ciliary maintenance (Figs. 4, 5, 6, and 8). Fmrp is a multifunctional RBP capable of translational repression[46,47] and important for mRNA delivery into neurites in the form of mRNP granules for local translation[2,21]. Our results suggest that Fmrp is involved in the formation and delivery of mRNP granules into ependymal cilia to provide mRNAs for the local translation (Fig. 8k). Firstly, the depletion of Fmrp repressed ciliary local translations, including the local synthesis of α-tubulin (Figs. 4e-i and 8i, j). If Fmrp functioned solely as a translation repressor, its depletion would have increased ciliary local translations. Nevertheless, as mRNAs in mRNP granules are translationally silenced during their delivery and storage, Fmrp might contribute to this translational incompetence till the mRNAs are released for local translation (Fig. 8k). Secondly, our RNA deep sequencing analyses suggest reduced abundance of many cilia-related mRNAs in FMRP-depleted ependymal cilia (Fig. 7a-c and Supplementary Table 1). Importantly, we confirmed that mRNAs of both α-tubulin and β-tubulin are indeed distributed in ependymal multicilia for the local translation of tubulin (Figs. 7d-g and 8), although other candidate mRNAs still require future validations. Thirdly, the requirement of ciliary localization of Fmrp to prevent ependymal multicilia from degeneration (Fig. 6) is also consistent with its proposed role in mRNA delivery, although our current results do not exclude the possibility that ciliary Fmrp might also have roles unrelated to local translation. Future studies will be required to identify other components of ciliary mRNP granules and understand their interplays with each other and with ciliary mRNAs. Interestingly, Cfap44, an inner dynein arm-tethering protein[57], has recently been reported to also bind RNA[58]. Upf1 and Caprin1 (Fig. 4b) could also be ciliary RBPs because they both have roles in neuronal local translation[41]. Our results suggest that ciliary mRNP granules are stored in the central lumen (Fig. 4c) and might release mRNAs for protein translation in response to certain signals or demands, analogous to what has been demonstrated by studies in neurites[4,59,60]. It will thus be interesting in the future to understand how the ciliary local translation is controlled to coordinate actions of the mRNA delivery machinery with the protein translation ones for proper maintenance of multicilia.

Local translation defects in cilia might not only contribute to ciliopathies but revise our knowledge in the pathology of other diseases as well. For instance, mutations in *FMR1*, mainly through trinucleotide expansion of CGG repeats in the 5′ untranslated region (UTR) and rarely through missense mutations in the coding region, is known to result in fragile X syndrome, the most common inherited form of mental retardation and single-gene cause of autism spectrum disorder, by dysregulating local translation in neurites critical for synaptic formation and functions[61,62]. The cerebrospinal fluid flow driven by ependymal cilia is important for homeostasis of the central nervous system[63,64]. Defects in ependymal multicilia can lead to hydrocephalus and damages in brain tissues and functions[9]. The Fmrp depletion-induced ependymal cilia degeneration (Figs. 5, 6) might thus also contribute to the pathology of fragile X syndrome[65]. Consistently, fragile X syndrome patients frequently display expanded brain ventricles[66,67] resembling hydrocephalus. In addition, *Fmr1*-deificent mice display age-dependent loss of primary cilia in neurons of the dentate gyrus (DG)[68]. Therefore, whether primary cilia possess local translation and contribute to the progression of fragile X syndrome is also an interesting question.

## Methods

**Plasmid construction**. For lentiviral expression of GFP-fusion proteins, the full-length cDNAs of mouse Rpl11 (NM_025919), Rps3 (NM_012052), Centrin1 (NM_007593), Eif3d (NM_018749), Eif3h (NM_080635), Eif3m (NM_145380), and Fmrp (NM_008031) were PCR-amplified from mEPC cDNAs and cloned into the pLV-EGFP-C1 vector.

To create Fmr-i1-insensitive Fmrp-expression adenovirus for rescue experiments, the Fmrp cDNA containing synonymous mutations (5′ CCACCTGT AGGTTATAATAAA 3′ → 5′ CCTCCAGTTGGATACAACAAG 3′) was generated by PCR, cloned into pYr1.1-EGFP to express GFP-Fmrp, and then recombined into pAd/Block-it-DEST by LR clonase II enzyme mix (ThermoFisher Scientific, 11791020). To tether GFP-Fmrp to the outer membrane of mitochondria, the C-terminal transmembrane domain of human monoamine oxidase (NM_000240, 481-528 aa)[49] was fused to the C-terminus of the siRNA-resistant Fmrp. Centrin1 cDNA was cloned into the pYr1.1-EGFP vector to express Centrin1-GFP.

Primers used for cloning were listed in Supplementary Table 2. All the plasmid constructs were verified by sequencing.

**Cell culture, transfection, and viral infection**. HEK293T (ATCC, CRL-11268) and HEK293A (ThermoFisher Scientific, R70507) cells were maintained in DMEM (Gibco, C11965500BT) supplemented with 10% fetal bovine serum (Ausbian, VS500T), 0.3 mg/ml L-glutamine (Sigma–Aldrich, G8540), 100 U/ml penicillin and 100 μg/ml streptomycin (Invitrogen, 15140122) at 37 °C in an atmosphere containing 5% $CO_2$. mEPCs were isolated and cultured as described[20,23,69]. The telencephalon from postnatal day 0 to 1 C57BL/6 mouse pups of either sex were dissected after removing the cerebellum, olfactory bulbs and hippocampus with tweezers (Dumont, 1214Y84) in cold dissection solution (161 mM NaCl, 5 mM KCl, 1 mM $MgSO_4$, 3.7 mM $CaCl_2$, 5 mM Hepes, and 5.5 mM Glucose, pH 7.4) under a stereo microscope. The telencephalon were digested with 1 ml of the dissection solution containing 10 U/ml papain (Worthington, LS003126), 0.2 mg/ml L-Cysteine, 0.5 mM EDTA, 1 mM $CaCl_2$, 1.5 mM NaOH, and 0.15% DNase I (Sigma, D5025) for 30 min at 37 °C. Cells were dissociated by pipetting up and down 10 times with a 5 ml pipette and collected by centrifugation at $400 \times g$ for 5 min at room temperature. Cells were resuspended with DMEM medium supplemented with 10% fetal bovine serum (FBS) and 50 μg/ml primocin (InvivoGen), and inoculated into the flask coated by 10 μg/ml fibronectin (Sigma, FC010). Neurons were shaken off and removed after culturing for 1 days after inoculation. The remaining cells were further cultured to ~80% confluency (usually 2 days after) and then transferred into fibronectin-coated 29 mm glass-bottom dishes (Cellvis, D29-14-1.5-N) for immunofluorescence staining or 75-cm² fibronectin-coated flasks for cilia purification. After cells were confluent, FBS was removed from the medium to initiate differentiation. Cultured cells were routinely tested for mycoplasma contamination. mEPCs grown in 29 mm glass-bottom dishes (Cellvis, D29-14-1.5-N) were transfected with 40 pmol siRNA and 3 μl Lipofectamine RNAiMAX (ThermoFisher Scientific, 13778-150) every 3 days from day 1 before serum starvation (day −1). Two independent siRNAs against *Fmr1* (Fmr-i1 and Fmr-i2) and the standard non-targeting siRNA (Ctrl-i) were used (GenePharma). The sequences of oligonucleotides are: Ctrl-i (negative control): 5′- TTCTCCGAACG TGTCACGTtt-3′; Fmr-i1: 5′-UUUAUUAUAACCUACAGGUtt-3′; Fmr-i2: 5′- UA AACCUCAACUUCAUCACtt -3′.

Lentivirus production and infection were performed as described[70]. To produce adenovirus, adenoviral plasmids were linearized with PacI and transfected into HEK293A cells with Lipofectmine 2000 (ThermoFisher Scientific, 11668-019). After ~80% cells showed a cytopathic effect, cells were harvested and virus were released by three freeze-thaw (−80 °C and 37 °C) cycles. The harvested adenoviral particles were used at a 1:1000 dilution for infection in mEPCs.

Experiments involving mouse tissues were performed in accordance with the ethical guidelines of Shanghai Institute of Biochemistry and Cell Biology, Chinese Academy of Sciences, and approved by the Institutional Animal Care and Use Committee.

**Immunoblotting**. For immunoblotting, samples were lysed in 2× SDS-loading buffer and boiled for 3 min at 95 °C. Proteins were separated by SDS-PAGE on Precast 4–12% Bis-Tris gels (Biofuraw, 180-8008H) and transferred onto nitrocellulose membrane with 0.2 μm pore size (GE Healthcare, 10600001). After three washes in TBST (150 mM NaCl, 50 mM Tris-HCl, 0.05% Tween 20, pH 7.5), the membrane was incubated in blocking buffer (5% skimmed milk powder in TBST) for 1 h, followed by incubation with primary antibodies in antibody dilution buffer (4% BSA in TBST) overnight at 4 °C. After three washes in TBST for 15 min each, the membrane was incubated with HRP-conjugated secondary antibodies in antibody dilution buffer for 1 h and washed for 15 min three times. Western Lightning Plus-ECL (PerkinElmer, NEL104001EA) was used to detect proteins on membranes, and the luminescent signals were captured with X-ray films (Fig. 1c)

or MiniChemi Chemiluminescence imager (Sagecreation) (Figs. 4f, 6c, 7b and Supplementary Fig. 1a). After the detection of IFT81 and RPL4 (Fig. 1c), the immunoblots were stripped off previous antibodies by incubating in stripping buffer (EpiZyme, PS107) for 15 min and used to detect eIF3d and eIF3h, respectively.

Antibodies for immunoblotting were listed in Supplementary Table 3.

**Fluorescent staining.** mEPCs cultured in 29 mm glass-bottom dishes (Cellvis, D29-14-1.5-N) were pre-extracted with 0.5% Triton X-100 in PBS for 30 s, followed by fixation with 4% fresh paraformaldehyde (PFA) in PBS for 15 min at room temperature. After fixation, the cells were permeabilized with 0.5% Triton X-100 in PBS for 15 min. To label multicilia-containing apical cortexes (Fig. 3e, f and d) and TZ-containing ciliary shafts (Figs. 3h, 8g and Supplementary Fig. 2c), the ciliary preparations were spun onto 12 mm glass coverslips coated with poly-L-lysine (Sigma–Aldrich, P1399) at $4200 \times g$ for 10 min at 4 °C, followed by the fixation and permeabilization. Immunofluorescent staining was carried out as described with some modifications[70]. The samples were incubated with blocking buffer (4% BSA in TBST) for an hour at room temperature, followed by incubation with primary antibodies in blocking buffer overnight at 4 °C. After three washes in blocking buffer for 5 min each, the samples were incubated with secondary antibodies in blocking buffer for an hour at room temperature. After three washes in blocking buffer for 5 min each, the samples were mounted with ProLong™ (ThermoFisher Scientific, 2273640).

Antibodies for immunofluorescent staining were listed in Supplementary Table 3.

**Acridine orange (AO) staining.** AO staining was performed as described[27], with some modifications. mEPCs cultured in 29 mm glass-bottom dishes (Cellvis, D29-14-1.5-N) to day 10 post serum starvation were incubated with 100 nM siR-tubulin (SPIROCHROME, CY-SC002) for 1 h at 37 °C, followed by fixation with 4% PFA in PBS for 15 min at 4 °C. After treated with PBS (mock) or 100 µg/ml RNase A (ThermoFisher Scientific, 12091021) in PBS for 30 min at 37 °C, the cells were incubated with 20 µg/ml AO (Sigma–Aldrich, 318337) in staining buffer (37 mM citric acid, 126 mM Na$_2$HPO$_4$, 150 mM NaCl, 1 mM EDTA, pH 3.8) for 15 min at 4 °C. After staining, the cells were mounted in Prolong Diamond Antifade Mountant (ThermoFisher Scientific, P36970) and immediately imaged on a Leica TCS SP8 WLL system with 561 excitation/ 650 emission (for RNA fluorescence), 488 excitation/ 525 emission (for DNA fluorescence), and 650 excitation/690 emission (for siR-tubulin fluorescence) filter sets. As siR-tubulin staining of multicilia tended to fade rapidly if permeabolized with Triton X-100, we chose to directly treat the cells with RNase without detergent permeabilization. We confirmed that the RNase treatment was effective (Supplementary Fig. 1b).

**Pyronin Y (PY)–Hoechst 33342 staining.** PY-Hoechst 33342 staining was performed as described[28]. mEPCs cultured in 29 mm glass-bottom dishes (Cellvis, D29-14-1.5-N) to day 10 were fixed with 4% PFA in PBS for 15 min, followed by permeabilization with 0.5% Triton X-100 in PBS for 15 min at 4 °C. After treated with PBS (mock) or 100 µg/ml RNase A (ThermoFisher Scientific, 12091021) in PBS for 30 min at 37 °C, the cells were incubated with 2 µg/ml PY (MedChemExpress, HY-D0971) and 4 µg/ml Hoechst 33342 (ThermoFisher Scientific, H3570) in PBS for 1 h at 4 °C. Then the cells were blocked with 2 µg/ml donkey normal IgG (Yeasen, 36108ES10) in PBS for 30 min and incubated with mouse anti-acetylated Tubulin (Sigma–Aldrich, T6793, 1:1000) for 1 h, followed by incubation with donkey anti-mouse IgG conjugated with Alexa Fluor 488 (ThermoFisher Scientific, A-21202, 1:1000). After washing, the cells were mounted in Prolong Diamond Antifade Mountant (ThermoFisher Scientific, P36970) and imaged immediately.

**Fluorescence in situ hybridization (FISH).** mEPCs cultured in 29 mm glass-bottom dishes (Cellvis, D29-14-1.5-N) to day 10 were fixed with 4% PFA in PBS for 15 min and washed in PBS for three times (5 min each). The cells were permeabilized in 75% ethanol overnight at 4 °C, then treated with PBS (mock) or 100 µg/ml RNase A (ThermoFisher Scientific, 12091021) for 30 min at 37 °C, and used for FISH as follows.

FISH with the Dig-labeled RNA probes were performed as described[30]. A gene fragment of the 18 S ribosomal RNA (NR_003278, 490–1489 nt) was PCR-amplified from mEPC genomic DNA and cloned into pCDNA3.0. After linearization, the plasmid was used as the template to transcribe DIG-rUTP-labeled RNA probes with the DIG RNA labeling kit (Roche, 11175025910). The antisense and sense probes, generated, respectively, through the T7 promoter and the SP6 promoter, were purified using MEGAclear Transcription Clean-Up Kit (ThermoFisher Scientific, AM1908). The mock- or RNase-treated mEPCs were washed with 50% formamide in 2× saline sodium citrate (SSC) for 10 min, followed by pre-hybridization in hybridization buffer (10% dextran sulfate, 50% formamide, 1 mg/ml yeast tRNA, 5 mM Ribonucleoside Vanadyl Complex in 1×SSC) at 50 °C for 2 h. The cells were then incubated with 10 ng/µl Dig-labeled RNA probes in hybridization buffer for 16 h at 50 °C, followed by two rounds of washes with 50% formamide in 2× SSC at 50 °C (20 min each) and one wash with PBS for 5 min. The hybridized mEPCs were blocked with 2 µg/ml donkey normal IgG (Yeasen, 36108ES10) in PBS for 30 min, followed by incubation with mouse anti-acetylated

Tubulin (Sigma–Aldrich, T6793, 1:1000) and sheep anti-digoxigenin (Roche, 11333089001, 1:200) overnight at 4 °C. After three washes with PBS (5 min each), the mEPCs were incubated with donkey anti-sheep IgG conjugated with Alexa Fluor 546 (ThermoFisher Scientific, A-21098, 1:1000) and donkey anti-mouse IgG conjugated with Alexa Fluor 488 (ThermoFisher Scientific, A-21202, 1:1000) for 1 h.

FISH with the single-molecule (sm) DNA probe sets were carried out as described[29]. Mixed DNA probes for the detection of the transcripts of *Tuba1a* (NM_011653), *Tuba1b* (NM_011654), and *Tuba1c* (NM_009448) or the transcripts of *Tubb2a* (NM_009450) and *Tubb2b* (NM_023716) were designed using the Stellaris® RNA FISH Probe Designer (www.biosearchtech.com/stellarisdesigner; Biosearch Technologies) and their sequences are listed in Supplementary Table 4. Probes against 18 S rRNA was from the positive control of Fluorescent In situ Hybridization Kit (RiboBio, C10910). The mock- or RNase-treated mEPCs were washed with 10% formamide in 2× SSC for 10 min, followed by pre-hybridization in hybridization buffer (10% formamide,10% dextran sulfate in 2 × SSC) at 37 °C for 1 h. Subsequently, the mEPCs were incubated with Cy3-labeled smFISH probes (RiboBio) in hybridization buffer for 16 h at 37 °C (100 nM probes for α-tubulin and β-tubulin mRNA, 500 nM probes for 18 S rRNA). After three washes with 10% formamide in 2× SSC (5 min each) at 37 °C. The follow-up immunofluorescent staining for acetylated Tubulin was carried out as described in the PY-Hoechst staining.

**Cilia isolation.** Cilia purification for mass spectrometry was performed as described in Fig. 1a[23]. To purify multicilia-containing apical cortexes from mEPCs, one 75 cm$^2$ flask of mEPCs at day 10 was washed twice with ice-cold PBS and twice with the Ca$^{2+}$-free deciliation buffer (20 mM Hepes, 25 mM KCl, 250 mM Sucrose, 1 mM EDTA, pH 7.5)[36]. Nine milliliters of deciliation buffer containing 0.01% Triton X-100 were added to the cells. The flask was then shaken horizontally at 300 rpm in a 37 °C shaker (Zhichu, ZQZY-AS9) for 30 min to release the ciliated apical cortexes. The suspension was collected and centrifuged at $600 \times g$ for 10 min at 4 °C. For immunofluorescent labeling (Fig. 3e) and puromycin labeling (Fig. 3f), the multicilia-containing pellet was resuspended with 1 ml of fresh culture medium. For ciliary RNA purification and deep sequencing (Fig. 7a), the pellet was resuspended with 400 µl of deciliation buffer and loaded on the top of a three-step discontinuous OptiPrep density gradient (Sigma–Aldrich, D1556, 60% in stock) (1 ml 25% iodixanol, 1.8 ml 20% iodixanol, and 1.8 ml 15% iodixanol in deciliation buffer from bottom to top). After centrifugation at $300,000 \times g$ for 18 h at 4 °C, multicilia-containing apical cortexes were collected from the interphase between the 15% and 20% iodixanol solution.

To isolated ciliary shafts from mEPCs, one 75-cm$^2$ flask of mEPCs at day 10 was washed twice with ice-cold PBS and twice with the Ca$^{2+}$-containing deciliation buffer (20 mM Pipes, 20 mM CaCl$_2$, 250 mM Sucrose, pH 5.5)[23]. 9 ml of deciliation buffer containing 0.01% Triton X-100 were added to the cells. The flask was then shaken horizontally at 300 rpm in a 37 °C shaker (Zhichu, ZQZY-AS9) for 15 min to detach ciliary shafts from their basal bodies. The suspension containing ciliary shafts was collected and centrifuged at $600 \times g$ for 10 min at 4 °C to remove pellets. For puromycin labeling (Fig. 3h) and Puro-PLA assay (Fig. 8g), ciliary shaft preparations were spun onto 12 mm glass coverslips coated with poly-L-lysine (Sigma–Aldrich, P1399) at $4200 \times g$ for 10 min at 4 °C.

**Puromycin labeling assay.** Puromycin labelling assay for the detection of nascent peptides was performed as described[32,33,51], with some modifications. Cultured mEPCs were treated with 10 µg/ml puromycin (Sigma–Aldrich, P8833) for 5 min or 10 min. To inhibit protein translation, mEPCs were treated with 10 µg/ml CHX (Sigma–Aldrich, C7698) for 30 min prior to the addition of puromycin. Puromycin-treated mEPCs were fixed in 4% PFA in PBS for 15 min and permeabilized with 0.5% Triton X-100 in PBS for 15 min. The cells were blocked with 4% BSA in TBST for 1 h, followed by incubation with mouse anti-puromycin (Merk Millipore, MABE343, 1:200) and rabbit anti-Rsph4a (1:400)[71] antibodies overnight at 4 °C. After three washes with the blocking buffer (5 min each), the cells were incubated with donkey anti-mouse IgG conjugated with Alexa Fluor 488 (ThermoFisher Scientific, A-21202, 1:1000) and donkey anti-rabbit IgG conjugated with Cy3 (Jackson ImmunoResearch, 711-165-152, 1:1000) for 1 h.

For puromycin labelling of multicilia-containing apical cortexes, the multicilia preparations resuspended in 1 ml of fresh culture medium (please refer to cilia purification) were incubated with 10 µg/ml puromycin (Sigma–Aldrich, P8833) for 30 min at 37 °C, or pre-treated with 10 µg/ml CHX for 30 min prior to the addition of puromycin. The samples were then centrifuged to poly-L-lysine-coated coverslips at $4200 \times g$ for 10 min at 4 °C, followed by fixation, permeabilization, and immunofluorescent staining.

For puromycin labelling of isolated ciliary shafts, the cilia preparations adhered onto coverslips were incubated with 10 µg/ml puromycin (Sigma–Aldrich, P8833) for 5 min, 10 min and 30 min at 37 °C with or without a pretreatment of 10 µg/ml CHX for 30 min prior to the addition of puromycin. The samples were then fixed and permeabilized, followed by immunofluorescent staining.

**Proximity ligation assay (PLA).** PLA was carried out to detect puromycylated α-tubulin peptides by following the Duolink PLA fluorescence protocol

(Sigma–Aldrich)[50]. mEPCs at day 10 (Fig. 8c, e, i and Supplementary Fig. 4b) or isolated ciliary shafts on coverslips (Fig. 8g) were treated with 10 μg/ml puromycin (Sigma–Aldrich, P8833) for 10 min or 30 min at 37 °C, respectively. The samples were fixed with 4% PFA in PBS for 15 min. After permeabilization with 0.5% Triton X-100 in PBS for 15 min, the samples were blocked with Duolink blocking solution for 1 h at 37 °C, followed by incubation overnight at 4 °C with Duolink Antibody Diluent containing mouse anti-puromycin (Merk Millipore, MABE343, 1:200), rabbit anti-α-tubF (Proteintech, 11224-1-AP, 1:800), rabbit anti-α-tubC (Abcam, ab15246, 1:100), rabbit anti-GapdhN (Abcam, ab181603, 1:200), rabbit anti-Tom20F (Proteintech, 11802-1-AP, 1:200) and guinea pig anti-Hydin antibody (1:200)[23]. After three washes with Wash buffer A (5 min each), the cells were incubated with Duolink Antibody Diluent containing rabbit PLA$^{plus}$ (Sigma–Aldrich, DUO92002, 1:5), mouse PLA$^{minus}$ (Sigma–Aldrich, DUO92004, 1:5), and donkey anti-guinea pig IgG conjugated with Alexa Fluor 488 (Jackson ImmunoResearch, 706-545-148, 1:500) for 1 h at 37 °C, followed by the ligation reaction for 30 min at 37 °C and amplification reaction for 100 min at 37 °C using Duolink Fluorescent Detection Reagent Red (Sigma–Aldrich, DUO92008). After washing, the cells were mounted with Duolink In Situ Mounting Medium with DAPI (Sigma–Aldrich, DUO82040) for imaging.

**Fluorescent microscopy**. Confocal microscopy images were taken on the Leica TCS SP8 WLL system with a ×63/1.4 oil immersion objective. Optical sections were captured at 0.5 μm intervals and z-stack images were obtained with maximum-intensity projections. 3D-SIM super-resolution images were acquired using a DeltaVision OMX SR Imaging System (GE Healthcare) equipped with 4 sCMOS (scientific Complementary metal–oxide–semiconductor) cameras (Pco.edge) and a PlanApo ×60/1.42 oil objective (Olympus). Immersion oil with refractive index of 1.518 (Cargille) was used to minimize spherical aberrations. Optical sections were acquired at 125 nm z-steps. Raw OMX data were analyzed in SoftWORX 7.0 software (GE Healthcare) with the following procedure: OMX SI Reconstructed using Channel-specific Wiener filter (0.002 for 488, 0.004 for 568, 0.002 for 647), OMX Align Image and Quick Projection (maximum-intensity projection).

As cell bodies are highly abundant in RNA and components of translational machineries, z-projected images from multicilia-containing z-stacks were presented to avoid the interference of fluorescent signals from cell bodies, unless the influence was neglectable or otherwise mentioned.

**Live cell imaging**. mEPCs were infected with adenovirus at two days before serum starvation (day −2) to express Centrin1-GFP as basal body marker and transfected with Ctrl-i or Fmr-i1 every three days from day −1. At day 7, the cells were incubated with 100 nM siR-tubulin for 1 h at 37 °C, followed by live cell imaging using a spinning disk confocal microscope (Olympus SpinSR10) equipped with a UPlanXApo ×60/1.42 oil objective, ORCA-Flash 4.0 V3 digital CMOS camera (Hamamatsu), OBIS solid state lasers (coherent, 488 nm, 10% laser power, 100 ms exposure; 650 nm, 10% laser power, 100 ms exposure), and an incubation chamber (37 °C, 5% $CO_2$, and 80% humidity). Z-stack sectioning was performed at 0.5 μm intervals to cover a depth of 15 μm that included fluorescent signals from ciliary tip to base). The images were recorded at 10-min intervals for 24 h. The time-lapse images were processed and analyzed with Cellsens Dimension (Olympus) and Fiji (ImageJ).

**RNA sequencing**. Total RNA was isolated from purified multicilia using TRIzol Reagent (ThermoFisher Scientific, 15596-018). 10 μg linear acrylamide (Yeasen, 10408ES03) and 15 μg GlycoBlue (ThermoFisher Scientific, AM9515) were added to facilitate RNA precipitation and recovery. Isolated RNA was quantified using the Agilent Bioanalyzer 2100 (Agilent Technologies). Total mRNAs were captured by Poly(A) Magnetic Isolation Module (NEB, E7490) for generation of cDNA libraries using NEBNext Ultra II Directional RNA Library Prep Kit for Illumina (NEB, E7760). cDNA Libraries from three replicates were multiplexed and loaded on an Illumina HiSeq X instrument according to manufacturer's instructions. Sequencing was carried out using a 2 ×150 bp paired-end configuration. The raw reads were filtered by Cutadapt 1.9.1 to remove adaptors and low-quality reads. Afterwards, the cleaned reads were mapped to the mouse reference genome (Ensemble GRCm38/mm10) using Hist 2.0.1. The FPKM values were calculated by Cuffdiff (v2.2.1) using default parameters[72]. The quality control, library preparation, and RNA sequencing were performed by GENEWIZ (Suzhou, China).

To identify candidate ciliary mRNAs delivered by Fmrp, 270 cilia-related genes were selected according to gene ontology annotations of the AmiGo 2 database. Filters were set as follows, key words: cilia; organism: Mus musculus; type: protein; evidence: experimental evidence in manual assertion; GO class: cillum. The deep sequencing results of cilia purified from Ctrl-i and Fmr-i1-treated mEPCs were subjected to the differential expression analysis.

**Statistics and reproducibility**. All experiments were performed at least three times (Fig. 1d-e, Fig. 2c-e, Fig. 4c-d, Fig. 7b, and Fig. 8b). Band intensities of immunoblots were quantified using Adobe Photoshop as described[73]. Fluorescent intensities in micrographs were measured using LAS X (Leica). To analyze the ciliary intensity of PLA signals (Fig. 8e, f, g, h and Supplementary Fig. 4b), PLA

signals were converted to PLA spots using the Spots model with the default settings in Imaris. In Fig. 8g-h, the number of PLA spots in the ciliary shaft were counted. In Fig. 8e-f and Supplementary Fig. 4b, the number of PLA spots in both the ciliary shaft and the soma were quantified. As the z-depth of soma regions varied among cells, PLA spots in the soma region were normalized to the value in 2000 μm³, which was close to the average volume covered by the confocal imaging. Quantification results are presented as mean ± SD. Two-tailed Student's $t$-test was performed using Graphpad Prism 8.3.0. Differences were considered significant when $P$ was <0.05.

**Reporting summary**. Further information on research design is available in the Nature Research Reporting Summary linked to this article.

## Data availability

The Ciliary RNA-Seq datasets generated and analyzed in this study have been deposited in the GEO (Gene Expression Omnibus) with the accession codes (GSE179935). Raw data of proteomic analysis results (Figs. 1b and 4a) have been deposited to the ProteomeXchange Consortium via the iProX partner repository[74] with the accession code (PXD029174). Source data, including uncropped immunoblots, are provided with this paper. Source data are provided with this paper.

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

## Acknowledgements

We thank Profs. Lan Bao, Hong Cheng, Jingyi Hui from Shanghai institute of Biochemistry and Cell Biology, and Prof. Yixian Zheng from Carnegie Institute for Science for valuable discussions, Dr. Bin Wang from Shanghai institute of Biochemistry and Cell Biology for advices on RNA extraction. Dr. Wenjuan Cai, Yufei Zhang, and Da Xu from Olympus Corporation for technical supports on live cell imaging, and institutional core facilities for cell biology and molecular biology for instrumental and technical supports. We thank Dr. Chao Peng and Yue Yin of the Mass Spectrometry System at the National Facility for Protein Science in Shanghai (NFPS), Zhangjiang Lab, Shanghai Advanced Research Institute, Chinese Academy of Sciences, for data collection and analysis. This work was supported by National Natural Science Foundation of China (31991192 for X.Z. and 31970652 for X.Y.), National Key R&D Program of China (2017YFA0503500 for X.Z. and X.Y.), and Chinese Academy of Sciences (XDB19020102 for X.Z.).

## Author contributions

X.Z. and X.Y. conceived and directed the project; K.H. performed major experiments; Y.C. purified cilia for mass spectrometry; X.Z., X.Y., and K.H. designed experiments, interpreted data, and wrote the paper.

## Competing interests

The authors declare no competing interests.

**Additional information**

