## [Peer Review File · Nature Communications]

REVIEWER COMMENTS

Reviewer #1 (Remarks to the Author):

The manuscript by Hao et al use imaging and biochemical experiments to describe the presence of ribosomal proteins, translation elongation factors and RNA in the cilia of mouse ependymal cells. Using puromycin as a metabolic label, the authors detect nascent protein in the cilia. FMRP was also detected in the cilia and knock-down experiments indicated enhanced protein synthesis following FMRP knockdown. In the field of ciliary biology, it is generally accepted that the proteins needed in cilia are transported via intraflagellar transport, not synthesized locally. As such, the ideas presented by the authors are exciting and represent a new way of thinking. On the other hand, it is also crucial that the data and interpretations are extremely solid. With this in mind, my criticisms and suggestions are below:

1. In the cilia purification experiments (figure 1), the authors should provide additional controls to show that they have de-enriched for nuclear markers, etc.

2. In most of the immunolabeling, nucleotide labeling and FISH experiments, the data are not always entirely convincing. For example the AO does not look convincing since the DNA signal is Iso sensitive to RNase and the tubulin signal also goes down. In the 18S RNA FISH experiment, it appears that there is very little signal present in the tubulin-labelled processes. In IG, it appears that just a single process is labelled.

3. All imaging experiments need to have analyses associated with them.

4. Re: the puromycin labelling experiment.
 - i) 10 min of labelling is too long- as puromycylated peptides could diffuse (or be transported) within this time (e.g. see Enam et al., 2020, eLIFE). A shorter labeling of 3-5 min would be better.
 - ii) the metabolic labelling time could be made irrelevant if the authors metabolically label isolated ciliar. It is unclear in the figures (and text) when and where the authors are working with intact cells vs. purified cilia. For example, in figure 3- are intact cells used in A-C- and then purified cilia? The puromycin labelling in figure 3F looks of very poor quality. The image is very blebby- ad it is unclear what is actually labelled by puromycin.

5. The imaging experiment show in figure 4C is the nicest in the whole paper- the cilia is well isolated and the image is well-resolved.

6. In the FMRP knock-down experiment- it was surprising that the loss of FMRP was associated with a reduction in protein synthesis- since most studies have identified FMRP as a translational repressor. In these studies, loss of FMRP is associated with an increase, rather than a decrease, in translation. The authors seem to have disregarded or missed this literature in their discussion.

7. Re: the FMRP rescue experiment. I'm not sure I understand the logic of this experiment. Are there no mitochondria in the cilia? The GFP-FMRP signal is also present in the cytoplasm of the cell- and hence the rescue could be from a non-cilial source of FMRP.

8. Lastly, it would seem that the key piece of missing data is the EM confirmation of ribosomes. This will be the piece of proof that most skeptics in the field would likely find the most convincing.

Reviewer #2 (Remarks to the Author):

In this manuscript, Hao and colleagues looked to answer a long awaiting puzzle in the ciliary proteome. Previous studies showed that multiple proteins related to the translation, such as ribosomal proteins, were identified as the ciliary proteome. However, because of their high abundance inside the cell, they were typically ignored as contaminants. Here the authors claimed that the actual local translation occurred inside cilia, based on the immunostaining and biochemical analysis. To confirm their hypothesis, they also confirmed that newly synthesized proteins are detected using puromycin incorporation. To understand the role of local translation, they reported the RNA-binding protein FMRP seems to be involved in the local translation in the ependymal cilia.

However, in my opinion, the current data provided by the authors were not enough to claim "the local translation" inside the ciliary structure. Here are some critical issues that should be addressed for the validation.

1. The authors presented the immunostaining data of multiple components of the ribosome and the translational machinery inside the cilia (Fig1 and Fig2). However, they are all independent observations. If they are truly functional, I think they should be co-localized. Although they were detected inside the structure, it would be hard to say that they are active if they were not co-localized.

2. The authors started with the ciliary proteome from LC/MS/MS data. They claimed that cilia have many proteins involved in the translational machinery. However, this experiment did not have control. Because their approach is a kind of 'enrichment' for the ciliary proteome, rather than direct labeling like proximity labeling, it is still possible that the proteins from the cytoplasm and other

locations could be detected here. It should be ruled out with proper control, and/or the authors discussed it.

3. The authors claimed that newly synthesized peptides were presented in the ciliary structure, based on puromycin incorporation. Although they argued that the puncta pattern is the evidence of local translation, I think it cannot completely rule out the case for the active transportation or diffusion from the basal body. It is hard to rule out the transportation scenario even with a short (10 min) reaction if we can observe similar signals in the basal body or apical surface.

4. They also sequenced transcriptome from cilia enriched samples (Fig 7A), but they were not fully discussed. Are they also increased in the ciliary proteome? The two experiments' overlap was not that high (30/121=20% & 30/147=25%). Are they related to their abundance? How many of them are already known to be located on the cilia? (both overlapped 30 genes and the union of 238 genes) If ciliary genes are enriched here, what is the difference between these "locally translated genes" compared to all known ciliary proteome? Regarding this data, the authors should also deposit their raw data (LC/MS/MS and RNA-seq) to the public repository, such as the PRIDE and the GEO databases.

5. They investigated FMRP as a candidate for the translational regulator in the cilia, based on the overlap of ciliary proteome and RNA-binding proteome (Fig 4). It is already known that RNA-binding protein may have a role in the ciliary function, like the Cfp44 case that is also referred to in this manuscript. Based on the data, I think FMRP protein and mRNA seem to have essential roles in maintaining the cilia function. However, I could not connect this with 'the local translation.' FMRP is known as the negative regulator of translation (Laggerbauer, HMG, 2001). And FMRP seems to be enriched in "the central lumen of axoneme (Fig. 4C)", but elongation factors seem to be located on "axonemal outer MT doublets (Fig. 2E)". How could they connect them?

6. They claimed that the tubulin is locally translated, based on proximity labeling assay with Anti-Puro and Anti-a-tubC (Fig 7). They showed no difference between Anti-a-tubF and Anti-a-tubC here. Still, they did not show the effect of Puro incorporation on the function of Anti-a-tubC. Suppose Puro incorporation hinders the activity of Anti-a-tubC. In that case, I suspect the result can be observed without the local translation. It should be clarified.

Reviewer #3 (Remarks to the Author):

The manuscript from Hao et al. in the Zhu laboratory provides evidence of local RNA translation within motile cilia of mouse multiciliated ependymal cells. Firstly, by using biochemical purification of motile cilia and mass spectrometry they detect enrichment of a large portion of ribosomal

proteins and eukaryotic initiation factors (eIFs). Subsequently, they show by immunofluorescence that some ribosomal proteins and eIFs components are found as puncta along the axonemes of motile cilia and that local translation is presumably happening within motile cilia by using a “Puromycin” translation initiation assay. Lastly, the authors further suggest a link between RNA translation in motile cilia to FMRP, a protein component of ribosomal RNA granules previously shown to be critical for RNA transport required for local translation. Loss of function experiment show that FMRP is required not for motile cilia formation, but for cilia “degeneration”. Using FISH the authors show presence of alpha-tubulin mRNA in motile cilia. Using PLA proximity-labeling assay, they show that local translation is happening in proximity of alpha-tubulin.

This is a very interesting and seemingly well-constructed manuscript that is employing a number of relatively complex methodologies to answer a long-standing question in motile cilia biology: is RNA translation locally achieved in the cilium, an organelle compartmentalized from the rest of the cytoplasm by a diffusion barrier present at the transition zone?

This manuscript is a potentially important paper in the cilia field that might be changing our view of translation in this organelle. In the current form there are some technical issues and conclusions not fully supported by the experimental data that precludes its publication in Nature Communications. This reviewer believes that an extensively revised manuscript, if the experimental data keep supporting the manuscript main conclusions, will be suitable for publication in Nature Communications.

A major issue in the manuscript is that the authors use two different protocols for purification of motile cilia, which lead to substantially different purified fractions, but which are used almost interchangeably in the text. Initially, to identify ribosomal and eIFs proteins in motile cilia of mouse ependymal cells, they use a protocol modified from the literature that leads to purification of axonemes, but not basal bodies by CaCl₂ treatment and centrifugation. From Fig. 3 onwards, they use a different protocol that involves using shearing force to strip motile cilia from the surface of ependymal cells as described in Anderson in JCB in 1974. As the authors state and show, this treatment leads to the stripping of the apical membrane region of ependymal cells, which also contains basal bodies and (presumably the transition zone) as shown in Fig. 3d,f by labeling through Odf2 staining. As shown in Fig.3f the puro incorporation signal, which detect newly translated peptides is located largely in the basal body region and not in the cilia region. Similarly, most of the RPs and eIF staining is localized in the same basal body region in Fig. 3e and also along the axoneme as also shown in Fig.1. Therefore, it remains unclear whether motile cilia local translation is happening mainly at the cortical region and rapidly diffusing in, instead of happening in the cilium proper (above the transition zone) as the authors claim. This is an important point as they author state in their main conclusions that there is local translation in motile cilia, an organelle biochemically separated from the rest of the cytoplasm by a gated region, the transition zone. To prove this point convincingly, they should 1. repeat the same experiments with motile cilia purified as in Fig1, without basal bodies and apical region, and 2. reduce the time of metabolic labeling to the minimum (below 10' if possible experimentally) to show that new peptides synthesis happens in motile cilia axonemes above the transition zone. It would be also important to show by imaging that

the transition zone remains somewhat intact after motile cilia purification as this remains an assumption at this stage.

The data shown in Fig.1 strongly suggest that ribosomal proteins are detected in the biochemically purified motile cilia fraction and are highly enriched in motile cilia of intact cells. Data in Figs. 1D-G seem to suggest that most of the translation machinery is concentrated in motile cilia at d10 of ependymal cells differentiation, which is somewhat surprising. It is important to validate this observation further by analyzing the enrichment of ribosomal and eIFs proteins after cells are stripped of cilia, that is compare the content of the motile cilia fraction to the deciliated fraction and not just to intact ependymal cells as done in Fig. 1C. As an aside, throughout the paper, I found difficult to fully assess images from cells, besides a few exceptions, as only the motile cilia region is shown.

The authors link FMRP function to motile cilia “degeneration” - maybe disassembly would be a more appropriate word to describe this phenomenon, unless the authors believe this is a new ciliary process - by performing well-controlled RNAi experiments and exciting imaging live cell imaging experiment in ependymal cells by labeling for 24 hrs with srTubulin.

In this section, the authors claim that FMRP is part of a ciliary mRNPs granule structure identified by deep RNA sequencing of semi-purified biochemically motile cilia responsible for local mRNA translation. As main experiments supporting this specific claim of a ciliary mRNP are done using mRNA deep sequencing data and show only the presence of other mRNPs mRNAs, I have a difficult time justifying the claim of the presence of a mRNPs granule structure in motile cilia. To suggest this conclusion, the authors should perform MS analysis of immunoprecipitated material from tagged-FMRP from ciliary fractions and identify other mRNPs components associated with FMRP.

While the current data show a clear role of FMRP in controlling motile cilia homeostasis, they only suggest, and not completely convincingly, that there might be a link between FMRP and motile cilia translation. Do the authors detect an increase in certain ciliary proteins identified by RNAseq, when they rescue cells depleted of FMRP by GFP-FMRP? This analysis would increase the confidence in the link between FMRP and translation, and not just degeneration of motile cilia.

What I also found unclear is that FMRP is localized in the central pair region similarly to RSPH4a, while the ribosomal proteins and eIFs are localized on the axonemal region, in a different location within motile cilia. How is local translation achieved if the components of the machinery are located in different places? Potential explanation would be useful to add to the manuscript.

In Figure 7 the authors use PLA assay to detect local peptide formation. As they detect proximity of molecules (in this case newly synthesized protein and tubulin protein), these experiments show no evidence of tubulin translation per se, but only proximity to tubulin, which is arguably quite abundant in motile cilia. Based on this view I found difficult to justify the absolute claim that specific tubulin translation is happening. To strengthen the findings, these experiments should be complemented by additional controls by adding anti-tubulin antibodies raised against NTD peptides

or other anti-Tubulin antibodies including ones specific for certain tubulin isoforms specifically expressed in motile cilia.

The data shown in Figure 7 show that there is no alpha-Tub mRNA in the ependymal cell interior, which I found is a bit surprising. It would be important to show the cells in their entirety to fully assess the data.

The deep sequencing experiments presented in Fig. 7a show the presence of several satellite and centrosomal proteins, which should not be part of the axonemal purified fraction. This might suggest that the motile cilia fraction utilized cannot be reliably used to identify enriched motile cilia mRNAs. Indeed deep sequencing has been performed on motile cilia purified with apical membranes (Fig.S3). A discussion is warranted.

Minor point:

It would be great to see supplemental material videos for the cilia degeneration experiments in the absence of FMRP.

It is unclear why different ribosomal proteins are analyzed in the western vs IF studies in Fig.1, some comment is required to help following the rationale.

Figure 1A. Spelling error mass spectrometry

Page 3 Line 39. Protruding

Page 3 Line 54. Sperm uses flagella for locomotion not motile cilia

Page 3 Line 57.manifests as a syndromic disease

Response to reviewers' comments

Reviewer #1 :

The manuscript by Hao et al use imaging and biochemical experiments to describe the presence of ribosomal proteins, translation elongation factors and RNA in the cilia of mouse ependymal cells. Using puromycin as a metabolic label, the authors detect nascent protein in the cilia. FMRP was also detected in the cilia and knock-down experiments indicated enhanced protein synthesis following FMRP knockdown. In the field of ciliary biology, it is generally accepted that the proteins needed in cilia are transported via intraflagellar transport, not synthesized locally. As such, the ideas presented by the authors are exciting and represent a new way of thinking. On the other hand, it is also crucial that the data and interpretations are extremely solid. With this in mind, my criticisms and suggestions are below:

Response:

We thank the reviewer for appreciating the idea of this study and for critical comments/suggestions that have helped us to substantially improve the manuscript. In the revised manuscript, we have provided solid evidence for ciliary local translation by demonstrating that isolated individual ciliary shafts actively synthesize proteins (including α -tubulin) (Figs 3h, i, 8g, h and Supplementary Fig. 2c,d). We have also presented results to show that cytoplasmic puromycylated peptides of the abundant cytosolic protein Gapdh and mitochondrial protein Tom20 do not substantially diffuse into ependymal cilia (Fig. 8e, f and Supplementary Fig. 4), thus allowing us to convincingly attribute the ciliary Puro IF signals in mEPCs and isolated multicilia (Fig. 3b, c, f and Supplementary Fig. 2a, b) to ciliary local translation. We also performed other requested experiments and have revised the manuscript accordingly.

To avoid possible confusion, we would also like to clarify two typos in the above comments of our reviewer: (1) we examined translation initiation factors but not translation elongation factors; and (2) we report decreased, not enhanced, protein synthesis following FMRP knockdown.

1. In the cilia purification experiments (figure 1), the authors should provide additional controls to show that they have de-enriched for nuclear markers, etc.

Response:

This is a routine method in our laboratory for ependymal cilia purification. We have previously shown in multiple publications that ependymal cilia purified in this way were free of detectable contaminants for the nucleus by using Lamin B1 as marker (Duan et al., 2021; Zheng et al., 2019; Zhu et al., 2019) and cis-Golgi by using GM130 as marker (Duan et al., 2021). Following the request, we performed an additional set of experiments and included Lamin B1 and Tom20 as nuclear and mitochondrial markers, respectively. Following the request of reviewer #3, we loaded equal amount of total proteins each lane. We have presented the results in Supplementary Figure 1a in the revised manuscript. The results in

Figure 1c and Supplementary 1a both show similar ciliary enrichment of RPL4, eIF3h, and eIF3d or eIF3f over non-ciliary proteins.

2. In most of the immunolabeling, nucleotide labeling and FISH experiments, the data are not always entirely convincing. For example the AO does not look convincing since the DNA signal is also sensitive to RNase and the tubulin signal also goes down. In the 18S RNA FISH experiment, it appears that there is very little signal present in the tubulin-labelled processes. In IG, it appears that just a single process is labelled.

Response:

As cell bodies are highly abundant in RNA and components of translational machineries, our image acquisitions mainly focused on multicilia-containing regions to avoid the interference of cell bodies, unless the influence was neglectable (e.g., in negative controls). In Figure 1F of the original manuscript, we present images containing multiple multiciliated mEPCs to emphasize the ciliary RNA signals of AO. As apical surfaces of mEPCs were not exactly in a flat plane, the z-stack images of multicilia just happened to pick up nuclear DNA fluorescence of some mEPCs. Reviewer Figure 1 below shows a larger field of the mock-treated mEPCs covering a portion of the image presented in Figure 1F (blue arrows point to multiciliated mEPCs in Fig. 1F). We can clearly see that the imaging reached nuclear positions of these cells but not many other cells. Therefore, the intensities of DNA fluorescence herein are related to nuclear positions of different mEPCs instead of the RNase treatment. Our reviewer might miss the results in Figure S1A (Supplementary Fig. 1b in the revised manuscript), which are presented to indicate that the RNase A-treatment had little impact on the nuclear DNA fluorescence of AO but dramatically reduced the RNA fluorescence in the cell bodies. These images were from z-stacks covering just the cell body regions of the same set of samples, acquired from different fields after the imaging of cilia-containing regions was finished.

Reviewer Fig. 1

As the AO staining was somehow not compatible with immunofluorescent staining, we tried to use siR-tub to rapidly label multicilia. siR-tub, however, is a dye for live imaging and usually fades away after fixation. As described in the Method section, we found that its fluorescence could be retained in PFA-fixed cells without detergent extraction and thus successfully used it to counter-stain multicilia in the experiments, though multicilia intensities might vary slightly in different fields and samples.

To avoid confusion, we have replaced the mock-treated image in Figure 1F with one containing mEPCs pointed by yellow arrowheads in Reviewer Figure 1 (Fig. 1f, revised manuscript; please note that we tilted the original image before cropping to include these cells). The nuclear DNA staining of this image looks comparable to the RNase-treated one (Fig. 1f, revised manuscript). We also quantified relative RNA fluorescent intensity in multicilia (red AO fluorescence/siR-tub) to confirm the RNase-induced reduction of ciliary RNA fluorescence and have presented the results in the revised manuscript (Fig. 1g). We have modified the main text and figure legend accordingly to increase the clarity of our presentation.

In response to the concern on the FISH image of 18S rRNA, we have included magnified insets to show the presence of FISH signals in multiple cilia (arrows; Supplementary Fig. 1f, revised manuscript). We have moved the FISH results to Supplementary Figure 1f because we performed single-molecule FISH (smFISH) during revision and have included the results in Figure 1h in the revised manuscript. We have also presented two z-projected images from cilia-containing region and cell body region, respectively, to make it clear why z-stacked images from cilia-containing regions are necessary for the demonstration of ciliary fluorescent signals. We also quantified relative ciliary 18S rRNA intensities and have presented the results in Figure 1i in the revised manuscript.

3. All imaging experiments need to have analyses associated with them.

Response:

We performed the requested statistical analyses and have included these results in the revised manuscript.

4. Re: the puromycin labelling experiment.

i) 10 min of labelling is too long- as puromycylated peptides could diffuse (or be transported) within this time (e.g. see Enam et al., 2020, eLIFE). A shorter labeling of 3-5 min would be better.

Response:

We appreciate the comment and suggestion. We performed puromycin labeling for 5 min in mEPCs and obtained similar results as the 10-min labeling. We have included the

results in the revised manuscript (Fig. 3b,c) and accordingly moved the results of 10-min labeling to Supplementary Figure 2a-b.

We have also included additional negative controls in our Puro-PLA experiments to show that puromycylated peptides of the abundant cytosolic protein Gapdh and mitochondrial protein Tom20 do not diffuse into cilia to generate strong ciliary PLA signals (Fig. 8e, f and Supplementary Fig. 4). Therefore, the ciliary compartment is much less accessible to cytoplasmic puromycylated peptides than compartments inside the cytoplasm such as the nucleus (Enam et al., 2020), probably due to the barrier function of ciliary transition zone (TZ) (Garcia-Gonzalo and Reiter, 2017; Goncalves and Pelletier, 2017). These results also allow us to convincingly attribute the ciliary Puro IF signals in mEPCs and isolated multicilia (Fig. 3b, c, f and Supplementary Fig. 2a, b) to ciliary local translation.

ii) the metabolic labelling time could be made irrelevant if the authors metabolically label isolated cilia. It is unclear in the figures (and text) when and where the authors are working with intact cells vs. purified cilia. For example, in figure 3- are intact cells used in A-C- and then purified cilia? The puromycin labelling in figure 3F looks of very poor quality. The image is very blebby- and it is unclear what is actually labelled by puromycin.

Response:

We are grateful to our reviewer for the suggestion. During the revision, we established a procedure to isolate individual ciliary shafts (Fig. 3g, revised manuscript) and verified that they lacked basal bodies (only 0.3% of them were positive for basal body marker Odf2) and mostly (76.5%) contained transition zone (TZ) (Supplementary Fig. 2c,d). To our excitement, when the isolated ciliary shafts were subjected to Puro labeling from 5 min to 30 min, they actively displayed clear time- and translation-dependent IF signals of Puro (Fig. 3h, i). Puro-PLA experiments further indicated that the isolated ciliary shafts are able to locally synthesize α -tubulin (Fig. 8g, h). These results provide solid evidence for the existence of active ciliary local protein synthesis and suggest that the cilia have a high protein synthesis capacity.

Figure 3A-C were indeed from intact mEPCs and Figure 3D-F from isolated cilia. To avoid confusion, in the revised manuscript we have indicated in image panels whether mEPCs or isolated cilia are presented.

The Puro signals in multicilia in Figure 3F (top panels, Fig. 3f in the revised manuscript) do exist but are overwhelmed by the strong signals at the basal body area. In the revised manuscript, we have improved our presentation by providing magnified insets containing multiple cilia.

5. The imaging experiment show in figure 4C is the nicest in the whole paper- the cilia is well isolated and the image is well-resolved.

Response:

We thank our reviewer for appreciating our efforts.

6. In the FMRP knock-down experiment- it was surprising that the loss of FMRP was associated with a reduction in protein synthesis- since most studies have identified FMRP as a translational repressor. In these studies, loss of FMRP is associated with an increase, rather than a decrease, in translation. The authors seem to have disregarded or missed this literature in their discussion.

Response:

We are sorry for having not clearly described the rationale of this part of experiments. Protein local translation requires proper delivery of mRNAs, which are usually stored and transported in the form of mRNP granules, to their target compartments. mRNAs stored in mRNP granules are usually translationally inactive. They are released from the mRNP granules for protein translation upon local demand. FMRP is a multifunctional RNA-binding protein capable of functioning as both a component of mRNP granules to deliver mRNAs for local translation and a translational repressor in neurons (Buxbaum et al., 2015; Darnell et al., 2011; Jung et al., 2012; Lagerbauer et al., 2001). Our results similarly suggest that FMRP is also a component of mRNP granules responsible for mRNA delivery into ependymal multicilia. In this context, the reduced ciliary local protein synthesis of FMRP-depleted mEPCs (Figs 4e-i and 8i, j) is attributed primarily to the lack of ciliary mRNAs. Our results are therefore consistent with the documented roles of FMRP.

In the revised manuscript, we have provided rationales for the experiments in Figures 4 and 5 and an illustration model in Figure 8k to increase the clarity of our presentation. We have also modified the Results and Discussion sections accordingly.

7. Re: the FMRP rescue experiment. I'm not sure I understand the logic of this experiment. Are there no mitochondria in the cilia? The GFP-FMRP signal is also present in the cytoplasm of the cell- and hence the rescue could be from a non-ciliary source of FMRP.

Response:

We are sorry for having not clearly described the rationale of the rescue experiment. The ciliary compartment is generally believed to exclude mitochondria. This is why we used mitochondrial protein Tom20 as a marker for the quality of our purified cilia (Supplementary Fig. 1a, revised manuscript) and mitochondria-tethered FMRP construct (mtFMRP) to discriminate the ciliary function of FMRP from the cytoplasmic function. We demonstrate that GFP-mtFMRP indeed lacked ciliary localization and also did not rescue the multicilia degeneration phenotype of endogenous FMRP-depleted mEPCs, therefore verifying that FMRP needs to localize into cilia to prevent ciliary degeneration. GFP-FMRP was used as a positive control. We have provided rationales for the rescue experiment in the revised manuscript to increase the clarity of our presentation.

8. Lastly, it would seem that the key piece of missing data is the EM confirmation of ribosomes. This will be the piece of proof that most skeptics in the field would likely find the most convincing.

Response:

We agree with our reviewer that visualization of ciliary ribosomes will provide a strong evidence to our study. Motile cilia are densely packed with protein complexes responsible for ciliary beat (including outer and inner dynein arms, radial spokes, and central-pair microtubule projections) and abundant in protein complexes of the intraflagellar transport (IFT) machinery (including IFT-A, IFT-B, BBSome, and cytoplasmic dynein 2), which bidirectionally transports ciliary components into and out of cilia (Ishikawa and Marshall, 2017; Ishikawa, 2017; Mourao et al., 2016). As the ribosome is below 30 nm in size, convincing EM confirmation of ciliary ribosomes is expected to be a technical challenge and will demand cutting-edge technologies, expertise, and laborious work. For instance, although the local protein synthesis in neurites was initially reported decades ago and has been well accepted in the field (Holt et al., 2019), direct, convincing visualization of their ribosomes has still not been achieved. Recent papers elucidating the presence of ribosomes in neurites still depend on indirect methods, e.g., through immunofluorescence of individual subunits and gold particles in immuno-EM (Hafner et al., 2019; Holt et al., 2019; Shigeoka et al., 2016). We have provided strong evidence for the presence of translational machineries (Figs 1-2), local protein synthesis (Figs 3 and 7), mRNP components and mRNAs (Figs 4 and 7). Importantly, the metabolic labeling and PLA results of isolated individual ciliary shafts in the revised manuscript (Figs 3h, i and 8g, h) provide solid evidence for the existence of active local protein synthesis in ependymal cilia. Therefore, we hope our reviewer would allow the challenge of EM confirmation of ciliary ribosomes to be left to future investigations. We have pointed out that “characterization of detailed components of ciliary translational machineries, direct visualization of their localizations, and elucidation of their functioning mechanisms still require future investigations” in the 1st paragraph of the Discussion section in the revised manuscript.

Reviewer #2:

In this manuscript, Hao and colleagues looked to answer a long awaiting puzzle in the ciliary proteome. Previous studies showed that multiple proteins related to the translation, such as ribosomal proteins, were identified as the ciliary proteome. However, because of their high abundance inside the cell, they were typically ignored as contaminants. Here the authors claimed that the actual local translation occurred inside cilia, based on the immunostaining and biochemical analysis. To confirm their hypothesis, they also confirmed that newly synthesized proteins are detected using puromycin incorporation. To understand the role of local translation, they reported the RNA-binding protein FMRP seems to be involved in the local translation in the ependymal cilia.

However, in my opinion, the current data provided by the authors were not enough to claim

"the local translation" inside the ciliary structure. Here are some critical issues that should be addressed for the validation.

Response:

We thank the reviewer for critical comments\suggestions that have helped us to substantially improve the manuscript. In the revised manuscript, we have provided solid evidence for ciliary local translation by demonstrating that isolated individual ciliary shafts actively synthesize proteins (including α -tubulin) (Figs 3h, i, 8g, h and Supplementary Fig. 2c, d). We have also presented results to show that cytoplasmic puromycylated peptides of the abundant cytosolic protein Gapdh and mitochondrial protein Tom20 do not substantially diffuse into cilia (Fig. 8e, f and Supplementary Fig. 4), thus allowing us to convincingly attribute the ciliary Puro IF signals in mEPCs and isolated multicilia (Fig. 3b, c, f and Supplementary Fig. 2a, b) to ciliary local translation. We also performed other requested experiments and have revised the manuscript accordingly.

1. The authors presented the immunostaining data of multiple components of the ribosome and the translational machinery inside the cilia (Fig1 and Fig2). However, they are all independent observations. If they are truly functional, I think they should be co-localized. Although they were detected inside the structure, it would be hard to say that they are active if they were not co-localized.

Response:

We appreciate the insightful comments of our reviewer. We fully agree that subunits of a complex must stay together to function. This is also the foundation of our study because it would be weird to presume that the ribosomal and eIF components resided in cilia as individual molecules. Conventional fluorescent imaging, however, is usually not an appropriate way for assessing components within protein complexes in vivo. For instance, the size of ribosome is below 30 nm in diameter and sizes of eIF complexes are much smaller. It is thus impossible to distinguish one complex from another in vivo due to the limit of optical resolution, unless they are sparsely distributed (in this case the low fluorescent intensity becomes an issue). Primary antibodies and secondary antibodies are each about 16 nm in length and can display steric hindrance to preclude multiplex fluorescent labeling of different subunits in the same complex. Furthermore, multiplex colocalization studies require antibodies generated from different host animals. Unfortunately, all our available antibodies against ribosomal and eIF components are raised from rabbit.

Despite the above concerns, we performed multiplex microscopy following the request of our reviewer. We directly labeled the eIF3m antibody with an Alex Fluor 555 antibody labeling kit (ThermoFisher A10470) and used it to co-immunostain with anti-RPL10A antibody. We also immunostained RPS3 in mEPCs expressing GFP-RPL11. As shown in Reviewer Figure 2 below, in both cases we observed partial overlap between red and green fluorescent signals. Colocalization analysis using Fiji software also indicates their

considerable “colocalization”. Nevertheless, as discussed above, the merge of two colors does not necessarily mean that the labelled proteins co-exist in a protein complex. We thus choose not to present these results in the manuscript.

Reviewer Fig. 2

Comparing to the issue of colocalization, local protein translation is a much convincing way to confirm the function of protein translational machineries in cilia and is also the major focus of the manuscript. Therefore, we performed further experiments during the revision to exclude the possibility of peptide diffusion in the Puro labeling experiments and have provided additional solid evidence in the revised manuscript. Firstly, we demonstrate that isolated individual ciliary shafts lacking the basal body but containing transition zone (TZ) are still able to actively translate proteins, including α -tubulin (Figs 3g-i, Fig. 8g, h and Supplementary Fig. 2c, d). These results provide solid evidence for the existence of active ciliary local protein synthesis and also suggest that the cilia have a high protein synthesis capacity. Secondly, we have included additional negative controls in our Puro-PLA experiments to show that puromycylated peptides of the abundant cytosolic protein Gapdh and mitochondrial protein Tom20 do not diffuse into cilia to generate strong ciliary PLA signals (Fig. 8e, f and Supplementary Fig. 4). These results allow us to convincingly attribute the ciliary Puro IF signals in mEPCs and isolated multicilia (Fig. 3b, c, f and Supplementary Fig. 2a, b) to ciliary local translation.

We thus hope that our reviewer agree that the detailed characterization of ciliary translational machineries can be left to future investigations. We have pointed out that “characterization of detailed components of ciliary translational machineries, direct visualization of their localizations, and elucidation of their functioning mechanisms still require future investigations” (1st paragraph of the Discussion section, revised manuscript).

2. The authors started with the ciliary proteome from LC/MS/MS data. They claimed that cilia have many proteins involved in the translational machinery. However, this experiment did not have control. Because their approach is a kind of 'enrichment' for the ciliary proteome, rather than direct labeling like proximity labeling, it is still possible that the proteins from the cytoplasm and other locations could be detected here. It should be ruled out with proper control, and/or the authors discussed it.

Response:

We agree with our reviewer that contamination is an important issue in proteomic studies. It is actually a common issue for MS results from both fractionation and proximity labeling approaches. Therefore, it is critical to confirm that the proteins of interest are indeed distributed and function in cilia. We confirm that multiple representative subunits of the ribosome and eIF complexes are indeed located in ependymal multicilia (Figs 1d, e, 2, and 3e, revised manuscript). We also demonstrate that the cilia can actively synthesize proteins (Figs 3 and 8), indicating the existence of ciliary translational machineries. As we have not verified the ciliary localization of every ribosomal and eIF components identified in our MS experiments, we have included a sentence in the 1st paragraph of Discussion to emphasize that the detailed ciliary components still require future investigations, following the request of our reviewer.

3. The authors claimed that newly synthesized peptides were presented in the ciliary structure, based on puromycin incorporation. Although they argued that the puncta pattern is the evidence of local translation, I think it cannot completely rule out the case for the active transportation or diffusion from the basal body. It is hard to rule out the transportation scenario even with a short (10 min) reaction if we can observe similar signals in the basal body or apical surface.

Response:

We thank our reviewer for this critical comment. During the revision, we performed puromycin labeling for 5 min in mEPCs and obtained similar results as the 10-min labeling. We have included the results in the revised manuscript (Fig. 3b, c) and accordingly moved the results of 10-min labeling to Supplementary Figure 2a-b.

We have also included additional negative controls in our Puro-PLA experiments to show that puromycylated peptides of the abundant cytosolic protein Gapdh and mitochondrial protein Tom20 do not diffuse into cilia to generate strong ciliary PLA signals (Fig. 8e, f and

Supplementary Fig. 4). Therefore, the ciliary compartment is much less accessible to cytoplasmic puromycylated peptides than compartments inside the cytoplasm such as the nucleus (Enam et al., 2020), probably due to the barrier function of ciliary transition zone (TZ) (Garcia-Gonzalo and Reiter, 2017; Goncalves and Pelletier, 2017). These results also allow us to convincingly attribute the ciliary Puro IF signals in mEPCs and isolated multicilia (Fig. 3b, c, f and Supplementary Fig. 2a, b) to ciliary local translation.

To further exclude the possibility of active transportation or diffusion from the basal body or the cytoplasm, we isolated individual ciliary shafts (Fig. 3g) and verified that they lacked basal bodies (only 0.3% of them were positive for basal body marker Odf2) and mostly (76.5%) contained the transition zone (TZ), the diffusion barrier of cilia (Supplementary Fig. 2c,d). When the isolated ciliary shafts were subjected to Puro labeling from 5 min to 30 min, they actively displayed clear time- and translation-dependent IF signals of Puro (Fig. 3h, i). Puro-PLA experiments further indicated that the isolated ciliary shafts are able to locally synthesize α -tubulin (Fig. 8g, h). These results provide solid evidence for the existence of active ciliary local protein synthesis and also suggest that the cilia have a high protein synthesis capacity.

4. They also sequenced transcriptome from cilia enriched samples (Fig 7A), but they were not fully discussed. Are they also increased in the ciliary proteome? The two experiments' overlap was not that high (30/121=20% & 30/147=25%). Are they related to their abundance? How many of them are already known to be located on the cilia? (both overlapped 30 genes and the union of 238 genes) If ciliary genes are enriched here, what is the difference between these "locally translated genes" compared to all known ciliary proteome? Regarding this data, the authors should also deposit their raw data (LC/MS/MS and RNA-seq) to the public repository, such as the PRIDE and the GEO databases.

Response:

Cilia have not been documented to contain mRNAs. Therefore, our goal of the deep sequencing was to find candidate mRNAs so that we could validate at least one ciliary mRNA to corroborate the concept of ciliary local translation. Ciliary mRNAs were expected to be low in abundance comparing with cytoplasmic mRNAs and susceptible to degradation during purification. Therefore, to compromise between the yield of cilia and the degradation risk of their mRNAs, we chose to purify multicilia instead of individual ciliary shafts. To reduce possible influence of contaminants, we compared RNA deep sequencing results of cilia purified respectively from ctrl-i- and Fmr-i1-treated mEPCs and chose cilia-related mRNAs downregulated in the Fmr-i1-treated samples and overlapped in two sets of independent results as candidate ciliary mRNAs. In this way, we successfully identified α/β -tubulin mRNAs as ciliary mRNAs (Fig. 7, revised manuscript) and further confirmed that ependymal cilia locally translate α -tubulin, but not non-ciliary proteins Gapdh and Tom20 (Fig. 8 and Supplementary Fig. 4, revised manuscript). Nevertheless, as other candidate mRNAs have not been experimentally validated, it is inappropriate to consider them as "locally translated

genes" for further analysis, including comparing them with genes of all known ciliary proteome.

In response to our reviewer's comments, we further analyzed our deep sequencing results. Due to the update of gene ontology annotations in the AmiGo 2 database, the numbers of downregulated cilia-related genes upon FMRP depletion in Exp. 1 and Exp. 2 became 89 and 118 (in contrast to 91 and 117 in the original manuscript), respectively, and the overlapping genes became 36 (in contrast to 30 in the original manuscript) (Fig. 7c and Supplementary Data 1, revised manuscript). The majority of these candidate ciliary mRNAs (63/89 and 92/118) encoded proteins in ciliary shafts according to gene ontology annotations of the AmiGo 2 database (Supplementary Data 1). In addition to mRNAs of α - and β -tubulin isoforms, the 36 overlapped candidate mRNAs included those of MT regulators (*Dcdc2a* and *Dcx*) (Grati et al., 2015; Moores et al., 2004), tubulin polyglutamylase and glycyase (*Till9* and *Till10*) (Ikegami and Setou, 2009; Konno et al., 2016), axonemal dynein subunits (*Dnah12* and *Dnali1*) (King, 2016), subunits of axonemal dynein-docking complexes (*Drc7* and *Ttc25*) (Morohoshi et al., 2020; Wallmeier et al., 2016), IFT subunits (*Bbs1*, *Ift27*, and *Ift52*) (Ishikawa and Marshall, 2017), other proteins involved in the structure and function of motile cilia (e.g., *Armc9*, *Cep131*, *Cfap69*, *Dnajb13*, and *Tmem17*) (Dong et al., 2018; El Khouri et al., 2016; Garcia-Gonzalo and Reiter, 2017; Latour et al., 2020; Zhao et al., 2021), and even FMRP (*Fmr1*) (Fig. 7c, and Supplementary Table 1, revised manuscript).

We have modified the text to include these information and discussed potential implications of these candidate mRNAs in the revised manuscript. We also point out that "future validation of *bona fide* ciliary mRNAs additional to those of tubulin and elucidation of their translational regulations and physiological importance will lead to comprehensive understandings on roles of the ciliary local translation" in the Discussion.

The Ciliary RNA-Seq datasets generated and analyzed during the current study have been deposited in the GEO (Gene Expression Omnibus) with the accession codes: GSE179935 (<https://www.ncbi.nlm.nih.gov/geo/query/acc.cgi?acc=GSE179935>). The MS results are provided in the source data of the manuscript.

5. They investigated FMRP as a candidate for the translational regulator in the cilia, based on the overlap of ciliary proteome and RNA-binding proteome (Fig 4). It is already known that RNA-binding protein may have a role in the ciliary function, like the *Cfap44* case that is also referred to in this manuscript. Based on the data, I think FMRP protein and mRNA seem to have essential roles in maintaining the cilia function. However, I could not connect this with 'the local translation.' FMRP is known as the negative regulator of translation (Laggerbauer, HMG, 2001).

Response:

We are sorry for having not clearly described the rationale of this part of experiments. Protein local translation requires proper delivery of mRNAs, which are usually stored and transported in the form of mRNP granules, to their target compartments. mRNAs stored in mRNP granules are usually translationally inactive. They are released from the mRNP granules for protein translation upon local demand. FMRP is a multifunctional RNA-binding protein capable of functioning as both a component of mRNP granules to deliver mRNAs for local translation and a translational repressor in neurons (Buxbaum et al., 2015; Darnell et al., 2011; Jung et al., 2012; Laggerbauer et al., 2001). Our results similarly suggest that FMRP is also a component of mRNP granules responsible for mRNA delivery into ependymal multicilia. In this context, the reduced ciliary local protein synthesis of FMRP-depleted mEPCs (Figs 4e-i and 8i, j) is attributed primarily to the lack of ciliary mRNAs. Our results are therefore consistent with the documented roles of FMRP.

In the revised manuscript, we have modified the Results section to provide rationales for the experiments in Figures 4 and 5 and an illustration model in Figure 8k to increase the clarity of our presentation. To avoid bias, we have also included the sentence “our current results do not exclude the possibility that ciliary FMRP might also have roles unrelated to local translation” in the 3rd paragraph of Discussion.

6. And FMRP seems to be enriched in "the central lumen of axoneme (Fig. 4C)", but elongation factors seem to be located on "axonemal outer MT doublets (Fig. 2E)". How could they connect them?

Response:

We speculate that mRNP granules might use the central lumen as a storage room for ciliary mRNAs and release the mRNAs for protein translation in response to certain signaling or demands, analogous to what has been demonstrated by studies in neurites (Hafner et al., 2019; Holt et al., 2019; Shigeoka et al., 2016). We have included the following sentences in the Discussion section of the revised manuscript: Our results suggest that ciliary mRNP granules are stored in the central lumen (Fig. 4c) and might release mRNAs for protein translation in response to certain signals or demands, analogous to what has been demonstrated by studies in neurites (Hafner et al., 2019; Holt et al., 2019; Shigeoka et al., 2016). It will thus be interesting in the future to understand how the ciliary local translation is controlled to coordinate actions of the mRNA delivery machinery with the protein translation ones for proper maintenance of multicilia.

7. They claimed that the tubulin is locally translated, based on proximity labeling assay with Anti-Puro and Anti- α -tubC (Fig 7). They showed no difference between Anti- α -tubF and Anti- α -tubC here. Still, they did not show the effect of Puro incorporation on the function of Anti- α -tubC. Suppose Puro incorporation hinders the activity of Anti- α -tubC. In that case, I suspect the result can be observed without the local translation. It should be clarified.

Response:

We are sorry for having not clearly described the rationale of this part of experiments. As cilia are abundant in microtubules, we used anti- α -tubC to assess whether the strong PLA signals from the anti- α -tubF and anti-Puro antibodies (Fig. 8c, d) were produced specifically, i.e., through the concomitant binding of anti- α -tubulin and anti-Puro antibodies to the same puromycylated α -tubulin peptides as illustrated in Figure 8a, or non-specifically through their separate binding to α -tubulin molecules in axonemal MTs and any puromycylated peptides. As puromycylated α -tubulin peptides were expected to mostly lack epitopes of anti- α -tubC antibody (please refer to reviewer Fig. 3A below), anti- α -tubC antibody would only be able to generate much weaker ciliary PLA signals than anti- α -tubF did if the PLA signals indeed required the concomitant binding of anti- α -tubulin and anti-Puro antibodies to the same puromycylated α -tubulin peptides as a prerequisite. On the contrary, if PLA signals could be produced non-specifically through the separate antibody binding, anti- α -tubC antibody and anti- α -tubF antibody should generate comparable PLA signals with anti-Puro antibody (reviewer Fig. 3B). The actual experimental results (Fig. 8c, d, revised manuscript) fit the scenario illustrated in reviewer Figure 3A, indicating that the ciliary PLA signals were indeed generated by puromycylated α -tubulin peptides. We have modified the text for this part of experiments in the revised manuscript to improve the clarity of our presentation.

Reviewer Fig. 3

As the PLA reactions are highly sensitive and capable of detecting a single puromycylated peptide by principle. In the revised manuscript, we have further clarified the source of puromycylated α -tubulin peptides that generated the ciliary PLA signals. As the abundant cytosolic protein Gapdh and mitochondrial protein Tom20 were rarely detected in ependymal cilia (Supplementary Fig. 4a; also see Fig. 1c and Supplementary Fig. 1a for immunoblotting) (Zhang et al., 2019; Zheng et al., 2019), we examined ciliary PLA signals of their puromycylated peptides by using antibodies against N-terminal Gapdh (anti-GapdhN) and full-length Tom20 (anti-Tom20F). We observed that neither anti-GapdhN antibody nor anti-Tom20F antibody produced ciliary PLA signals comparable to anti- α -tubF antibody in parallel experiments (Fig. 8e, f and Supplementary Fig. 4b). We also performed PLA assays using isolated ciliary shafts and obtained similar results (Fig. 8g, h). These results provide

solid evidence to support that ependymal multicilia locally translate α -tubulin. They also indicate that the ciliary compartment is much less accessible to cytoplasmic puromycylated peptides than compartments inside the cytoplasm such as the nucleus (Enam et al., 2020), probably due to the barrier function of TZ (Garcia-Gonzalo and Reiter, 2017; Goncalves and Pelletier, 2017), thus allowing us to convincingly attribute the ciliary Puro IF signals in mEPCs and isolated multicilia (Fig. 3b, c, f and Supplementary Fig. 2a, b) to ciliary local translation.

Reviewer #3:

The manuscript from Hao et al. in the Zhu laboratory provides evidence of local RNA translation within motile cilia of mouse multiciliated ependymal cells. Firstly, by using biochemical purification of motile cilia and mass spectrometry they detect enrichment of a large portion of ribosomal proteins and eukaryotic initiation factors (eIFs). Subsequently, they show by immunofluorescence that some ribosomal proteins and eIFs components are found as puncta along the axonemes of motile cilia and that local translation is presumably happening within motile cilia by using a “Puromycin” translation initiation assay. Lastly, the authors further suggest a link between RNA translation in motile cilia to FMRP, a protein component of ribosomal RNA granules previously shown to be critical for RNA transport required for local translation. Loss of function experiment show that FMRP is required not for motile cilia formation, but for cilia “degeneration”. Using FISH the authors show presence of alpha-tubulin mRNA in motile cilia. Using PLA proximity-labeling assay, they show that local translation is happening in proximity of alpha-tubulin.

This is a very interesting and seemingly well-constructed manuscript that is employing a number of relatively complex methodologies to answer a long-standing question in motile cilia biology: is RNA translation locally achieved in the cilium, an organelle compartmentalized from the rest of the cytoplasm by a diffusion barrier present at the transition zone?

This manuscript is a potentially important paper in the cilia field that might be changing our view of translation in this organelle. In the current form there are some technical issues and conclusions not fully supported by the experimental data that precludes its publication in Nature Communications. This reviewer believes that an extensively revised manuscript, if the experimental data keep supporting the manuscript main conclusions, will be suitable for publication in Nature Communications.

Response:

We thank the reviewer for appreciating the idea of this study and for critical comments/suggestions that have helped us to improve the manuscript. In the revised manuscript, we have provided solid evidence for ciliary local translation by demonstrating that isolated individual ciliary shafts actively synthesize proteins (including α -tubulin) (Figs 3h, i, 8g, h and Supplementary Fig. 2c,d). We have also presented results to show that cytoplasmic puromycylated peptides of the abundant cytosolic protein Gapdh and

mitochondrial protein Tom20 do not substantially diffuse into ependymal cilia (Fig. 8e, f and Supplementary Fig. 4), thus allowing us to convincingly attribute the ciliary Puro IF signals in mEPCs and isolated multicilia (Fig. 3b, c, f and Supplementary Fig. 2a, b) to ciliary local translation. We also performed other requested experiments and have revised the manuscript accordingly.

1. A major issue in the manuscript is that the authors use two different protocols for purification of motile cilia, which lead to substantially different purified fractions, but which are used almost interchangeably in the text. Initially, to identify ribosomal and eIFs proteins in motile cilia of mouse ependymal cells, they use a protocol modified from the literature that leads to purification of axoneme, but not basal bodies by CaCl_2 treatment and centrifugation. From Fig. 3 onwards, they use a different protocol that involves using shearing force to strip motile cilia from the surface of ependymal cells as described in Anderson in JCB in 1974. As the authors state and show, this treatment leads to the stripping of the apical membrane region of ependymal cells, which also contains basal bodies and (presumably the transition zone) as shown in Fig. 3d,f by labeling through Odf2 staining. As shown in Fig.3f the puro incorporation signal, which detect newly translated peptides is located largely in the basal body region and not in the cilia region. Similarly, most of the RPs and eIF staining is localized in the same basal body region in Fig. 3e and also along the axoneme as also shown in Fig.1.

Therefore, it remains unclear whether motile cilia local translation is happening mainly at the cortical region and rapidly diffusing in, instead of happening in the cilium proper (above the transition zone) as the authors claim. This is an important point as they author state in their main conclusions that there is local translation in motile cilia, an organelle biochemically separated from the rest of the cytoplasm by a gated region, the transition zone. To prove this point convincingly, they should

i. Repeat the same experiments with motile cilia purified as in Fig1, without basal bodies and apical region.

Response:

We thank our reviewer for the clear dissection of our approaches and critical comments. We agree that our results presented in the original manuscript are unable to thoroughly exclude the possibility of active transportation or diffusion from the basal body and the cytoplasm. Following the request of our reviewer, we explored whether isolated ciliary shafts lacking the basal body but containing the transition zone (TZ) could display active protein translation. We did not use the protocol illustrated in Figure 1a, because vortexing suspended mEPCs would break the ciliary shafts and might cause the leakage, even loss, of ciliary contents important for protein synthesis. Instead, we added the Ca^{2+} -containing deciliation buffer directly to intact mEPCs in culture flasks and used the shearing force of a shaker to detach ciliary shafts from their cell bodies (Fig. 3g, revised manuscript). Immunostaining confirmed that the majority (76.5%) of ciliary shafts isolated in this way were positive for transition zone (TZ) marker Cep290 and lacked basal body marker Odf2, indicating that they contained TZ and lacked basal bodies. 23.2% of them were negative for

both markers. Only a very small portion (0.3%) of them were positive for both markers (Supplementary Fig. 2c,d). When the isolated ciliary shafts were subjected to Puro labeling from 5 min to 30 min, they actively displayed clear time- and translation-dependent IF signals of Puro (Fig. 3h, i). Puro-PLA experiments further indicated that the isolated ciliary shafts are able to locally synthesize α -tubulin (Fig. 8g, h). These results provide solid evidence for the existence of active ciliary local protein synthesis and suggest that the cilia have a high protein synthesis capacity. We are grateful to our reviewer for suggesting such an experiment.

ii. Reduce the time of metabolic labeling to the minimum (below 10' if possible experimentally) to show that new peptides synthesis happens in motile cilia axoneme above the transition zone.

Response:

We thank our reviewer for the suggestion. We performed puromycin labeling for 5 min in mEPCs during the revision and obtained similar results as the 10-min labeling. IF signals of Puro were readily detected along ciliary axonemes in the absence of CHX. We have included the results in the revised manuscript (Fig. 3b,c) and accordingly moved the results of 10-min labeling to Supplementary Figure 2a-b.

iii. It would be also important to show by imaging that the transition zone remains somewhat intact after motile cilia purification as this remains an assumption at this stage.

Response:

We are grateful to our reviewer for the clear instruction. As described above, we performed immunostaining to assess the quality of isolated ciliary shafts by using Cep290 and Odf2 as TZ and basal body markers, respectively. The majority (76.5%) of the isolated ciliary shafts contained TZ and lacked the basal body, 23.2% of them were negative for both markers, and only 0.3% of them were positive for both Cep290 and Odf2 (Supplementary Fig. 2c,d).

2. The data shown in Fig. 1 strongly suggest that ribosomal proteins are detected in the biochemically purified motile cilia fraction and are highly enriched in motile cilia of intact cells. Data in Figs. 1D-G seem to suggest that most of the translation machinery is concentrated in motile cilia at d10 of ependymal cells differentiation, which is somewhat surprising.

It is important to validate this observation further by analyzing the enrichment of ribosomal and eIFs proteins after cells are stripped of cilia, that is compare the content of the motile cilia fraction to the deciliated fraction and not just to intact ependymal cells as done in Fig. 1C. As an aside, throughout the paper, I found difficult to fully assess images from cells, besides a few exceptions, as only the motile cilia region is shown.

Response:

We chose to use mEPCs at d10 simply because multiciliated cells are relatively abundant (Delgehr et al., 2015), typically around 50% in our hands. We have clarified this in the revised manuscript (legend for Figure 1a). In our lab, mEPCs at day 7 and day 10 are routinely used in ciliary studies. Laboratory members just stick to a time for consistency (Zhao et al., 2021; Zheng et al., 2019). Therefore, our results do not implicate that the ciliary translation machinery is more abundant at d10.

As requested, we compared protein levels in intact mEPCs, deciliated somas, and purified cilia and have presented the results in the revised manuscript (Supplementary Fig. 1a). As expected, while typical ciliary proteins Rsph4a and IFT81 were highly enriched in the ciliary fraction, RPL4, eIF3f and eIF3h were moderately enriched comparing to Gapdh, Tom20 (a mitochondrial protein), and Lamin B1 (a nuclear protein). Quantification indicates that RPL4, eIF3f, and eIF3h were enriched by >6-fold over Tom20 (eIF3f/Tom20), similar to the results presented in Figure 1c, in which the enrichment was >7.5-fold over Gapdh.

As cell bodies are highly abundant in RNA and components of translational machineries, our image acquisitions mainly focused on multicilia-containing regions to avoid the interference of cell bodies, unless the influence was neglectable (e.g., in negative controls). In the revised manuscript, we have provided examples to show why this is necessary (Figs 1h, 7d, f; please see the images of the mock-treated samples).

3. The authors link FMRP function to motile cilia “degeneration” – maybe disassembly would be a more appropriate word to describe this phenomenon, unless the authors believe this is a new ciliary process - by performing well-controlled RNAi experiments and exciting imaging live cell imaging experiment in ependymal cells by labeling for 24 hrs with sirTubulin.

Response:

We thank our reviewer for appreciating our efforts and the comment. We agree that the cilia were certainly shortened through disassembly. Primary cilia are known to be disassembled gradually when cells enter the cell cycle. mEPCs, however, are terminally differentiated cells and their multicilia are not supposed to undergo physiological disassembly. Therefore, we feel that “degeneration” is a more accurate term to implicate the pathological nature of the phenotype. We have modified the text describing the results of Figure 5 to increase the clarity of presentation.

4. In this section, the authors claim that FMRP is part of a ciliary mRNPs granule structure identified by deep RNA sequencing of semi-purified biochemically motile cilia responsible for local mRNA translation. As main experiments supporting this specific claim of a ciliary mRNP are done using mRNA deep sequencing data and show only the presence of other mRNPs mRNAs, I have a difficult time justifying the claim of the presence of a mRNPs granule structure in motile cilia.

To suggest this conclusion, the authors should perform MS analysis of immunoprecipitated material from tagged-FMRP from ciliary fractions and identify other mRNPs components associated with FMRP.

Response:

We are sorry for having not clearly described the rationale of this part of experiments. Protein local translation requires proper delivery of mRNAs, which are usually stored and transported in the form of mRNP granules, to their target compartments. They are released from the mRNP granules for protein translation upon local demand. FMRP is a multifunctional RNA-binding protein capable of functioning as a component of mRNP granules to deliver mRNAs for local translation in neurites (Buxbaum et al., 2015; Jung et al., 2012; Laggerbauer et al., 2001). Similarly, ciliary local translation would also require mRNP granules to be transported into the ciliary shafts. Similar to the identification of possible components of ciliary translational machineries (Figs 1-2), we initially compared the proteome of our purified ependymal cilia (not deep RNA sequencing results) with an RNA-binding protein (RBP) database for candidate RBPs of ciliary mRNP granules for candidate RBPs (Fig. 4a), followed by analyzing their gene expressions during the multiciliation of cultured mouse tracheal epithelial cells to further narrow down to 6 candidate RBPs (Fig. 4b), among which FMRP is a well-known component of mRNP granules responsible for the mRNA delivery in local translation. We thus chose to focus on FMRP and confirmed that FMRP indeed shows ciliary distribution (Fig. 4a-g). Furthermore, its depletion-induced inhibition of ciliary local protein translation (Figs 4e-i and 8i, j), the importance of ciliary FMRP in multicilia maintenance (Figs 5, 6), and the presence of tubulin mRNA and local translation in cilia (Figs 7 and 8) are all consistent with the proposed role of ciliary mRNA delivery (Fig. 8k).

We thank our reviewer for the suggested approach to the identification of FMRP-associated proteins through MS analysis of immunoprecipitated material from ciliary fractions. mEPC culture requires the isolation of radial glial cells from neonatal mice (Delgehyr et al., 2015). We have not been successful in such IP-MS experiments due to the difficulty to obtain enough mEPCs for the cilia purification. Purified cilia are only sufficient for a couple of SDS-PAGE gels for immunoblotting. This is why we generally use ciliary proteome in this and other research projects (Duan et al., 2021; Zhang et al., 2019; Zheng et al., 2019). As this manuscript is aimed to demonstrate the concept of ciliary local translation, we hope our reviewer would agree that the identification and characterization of detailed components of ciliary mRNP granules can be left to future researches.

In the revised manuscript, we have provided rationales and modified the text for Figures 4 and 5 to increase the clarity of our presentation.

5. While the current data show a clear role of FMRP in controlling motile cilia homeostasis, they only suggest, and not completely convincingly, that there might be a link between FMRP and motile cilia translation.

i. Do the authors detect an increase in certain ciliary proteins identified by RNAseq, when they rescue cells depleted of FMRP by GFP-FMRP? This analysis would increase the confidence in the link between FMRP and translation, and not just degeneration of motile cilia.

Response:

As described above, FMRP was expected to be involved in ciliary delivery and storage of mRNAs in the form of mRNP granules. In the manuscript, we have provided multiple lines of evidence to link FMRP to the ciliary local translation: (1) The depletion of FMRP attenuated ciliary local protein synthesis (Fig. 4e-i); (2) We successfully identified α/β -tubulin mRNAs as *bona fide* ciliary mRNAs from a pool of candidate mRNAs whose abundance reduced in purified cilia of FMRP-depleted mEPCs based on RNA deep sequencing results (Fig. 7); (3) Ependymal cilia locally translated α -tubulin (Fig. 8a-h) and the local translation also required FMRP (Fig. 8i, j).

As the link between FMRP and local translation is well documented in neurons (Buxbaum et al., 2015; Jung et al., 2012; Lagerbauer et al., 2001), we did not spend our limited revision time to provide more supporting evidence on this by looking at the rescued cells. Instead, we put our major efforts to verify the existence of ciliary local translation, the most critical concern of our reviewers on the manuscript. In the revised manuscript, we have demonstrated that isolated individual ciliary shafts still actively synthesize proteins (including α -tubulin) (Figs 3h, i, 8g, h and Supplementary Fig. 2c,d). These results provide solid evidence for the existence of active local protein synthesis in ependymal cilia and also suggest that the cilia have a high protein synthesis capacity. Furthermore, we have also included additional negative controls in our Puro-PLA experiments to show that puromycylated peptides of the abundant cytosolic protein Gapdh and mitochondrial protein Tom20 do not substantially diffuse into cilia (Fig. 8e, f and Supplementary Fig. 4). These results indicate that the ciliary compartment is much less accessible to cytoplasmic puromycylated peptides than compartments inside the cytoplasm such as the nucleus (Enam et al., 2020), probably due to the barrier function of ciliary transition zone (TZ) (Garcia-Gonzalo and Reiter, 2017; Goncalves and Pelletier, 2017). These results allow us to convincingly attribute the ciliary Puro IF signals in mEPCs and isolated multicilia (Fig. 3b, c, f and Supplementary Fig. 2a, b) to ciliary local translation. We also performed quantifications and have included quantification results in the revised manuscript.

ii. What I also found unclear is that FMRP is localized in the central pair region similarly to RSPH4a, while the ribosomal proteins and eIFs are localized on the axonemal region, in a different location within motile cilia. How is local translation achieved if the components of the machinery are located in different places? Potential explanation would be useful to add to the manuscript.

Response:

We thank our reviewer for the comments. Local translation-related FMRP mRNP granules are usually for mRNA transport to and storage in target compartments. We speculate that mRNP granules might use the central lumen as a storage room for ciliary mRNAs and release the mRNAs for protein translation in response to certain signaling or demands, analogous to what has been demonstrated by studies in neurites (Hafner et al., 2019; Holt et al., 2019; Shigeoka et al., 2016). We have included the following sentences in the Discussion section of the revised manuscript: Our results suggest that ciliary mRNP granules are stored in the central lumen (Fig. 4c) and might release mRNAs for protein translation in response to certain signals or demands, analogous to what has been demonstrated by studies in neurites (Hafner et al., 2019; Holt et al., 2019; Shigeoka et al., 2016). It will thus be interesting in the future to understand how the ciliary local translation is controlled to coordinate actions of the mRNA delivery machinery with the protein translation ones for proper maintenance of multicilia.

6. In Figure 7 the authors use PLA assay to detect local peptide formation. As they detect proximity of molecules (in this case newly synthesized protein and tubulin protein), these experiments show no evidence of tubulin translation per se, but only proximity to tubulin, which is arguably quite abundant in motile cilia. Based on this view I found difficult to justify the absolute claim that specific tubulin translation is happening. To strengthen the findings, these experiments should be complemented by additional controls by adding anti-tubulin antibodies raised against NTD peptides or other anti-Tubulin antibodies including ones specific for certain tubulin isoforms specifically expressed in motile cilia.

Response:

We are sorry for having not clearly described the rationale of this part of experiments. We already clarify that the ciliary PLA signals were generated specially through the concomitant binding of anti- α -tubF and anti-Puro antibodies to the same puromycylated α -tubulin peptides by using anti-tubC antibody as a negative control. As puromycylated α -tubulin peptides were expected to mostly lack epitopes of anti- α -tubC antibody (please refer to reviewer Fig. 3A below), anti- α -tubC antibody would only be able to generate much weaker ciliary PLA signals than anti- α -tubF did if the PLA signals indeed required the concomitant binding of anti- α -tubulin and anti-Puro antibodies to the same puromycylated α -tubulin peptides (Fig. 8a) as a prerequisite. On the contrary, if PLA signals could be produced non-specifically, i.e., through the separate binding of anti- α -tubulin antibody to α -tubulin molecules in axonemal MTs and anti-Puro antibody to any puromycylated peptides, anti- α -tubC antibody and anti- α -tubF antibody should generate comparable PLA signals with anti-Puro antibody (please refer reviewer Fig. 3B). The actual experimental results (Fig. 8c, d, revised manuscript) fit the scenario illustrated in reviewer Figure 3A, indicating that the ciliary PLA signals were indeed generated by puromycylated α -tubulin peptides. We have modified the text for this part of experiments in the revised manuscript to improve the clarity of our presentation.

As the PLA reactions are highly sensitive and capable of detecting a single puromycylated peptide by principle, we have further clarified the source of puromycylated α -tubulin peptides that generated the ciliary PLA signals in the revised manuscript. As the abundant cytosolic protein Gapdh and mitochondrial protein Tom20 were rarely detected in ependymal cilia (Supplementary Fig. 4a; also see Fig. 1c and Supplementary Fig. 1a for immunoblotting) (Zhang et al., 2019; Zheng et al., 2019), we examined ciliary PLA signals of their puromycylated peptides by using antibodies against N-terminal Gapdh (anti-GapdhN) and full-length Tom20 (anti-Tom20F). We observed that neither anti-GapdhN antibody nor anti-Tom20F antibody produced ciliary PLA signals comparable to anti- α -tubF antibody in parallel experiments (Fig. 8e, f and Supplementary Fig. 4b). We also performed PLA assays using isolated ciliary shafts and obtained similar results (Fig. 8g, h). These results provide solid evidence for the local translation of α -tubulin in ependymal cilia. They also indicate that the ciliary compartment is much less accessible to cytoplasmic puromycylated peptides than compartments inside the cytoplasm such as the nucleus (Enam et al., 2020), probably due to the barrier function of TZ (Garcia-Gonzalo and Reiter, 2017; Goncalves and Pelletier, 2017), thus allowing us to convincingly attribute the ciliary Puro IF signals in mEPCs and isolated multicilia (Fig. 3b, c, f and Supplementary Fig. 2a, b) to ciliary local translation.

7. The data shown in Figure 7 show that there is no alpha-Tub mRNA in the ependymal cell interior, which I found is a bit surprising. It would be important to show the cells in their entirety to fully assess the data.

Response:

We are sorry for the confusion. As cell bodies are highly abundant in RNA (and also components of translational machineries and FMRP), our image acquisitions mainly focused on multicilia-containing regions to avoid the interference of cell bodies, unless the influence was neglectable (e.g., in negative controls). Following the request of our reviewer, we have provided images to show FISH signals in both cilia and cell bodies (Fig. 7d, f, revised manuscript; please see the images of the mock-treated samples).

8. The deep sequencing experiments presented in Fig. 7a show the presence of several satellite and centrosomal proteins, which should not be part of the axonemal purified fraction. This might suggest that the motile cilia fraction utilized cannot be reliably used to identify enriched motile cilia mRNAs. Indeed deep sequencing has been performed on motile cilia purified with apical membranes (Fig.S3). A discussion is warranted.

Response:

As ciliary mRNAs were expected to be low in abundance comparing with cytoplasmic mRNAs and susceptible to degradation during purification, we chose to purify multicilia instead of individual ciliary shafts as a compromise between the yield of cilia and the degradation risk of their mRNAs. We selected mRNAs whose protein localizations are related to centriole/basal body and ciliary shaft as candidate mRNAs because some centriole-related proteins might be found to localize in motile cilia as well. For instance, Cep131 is a centriole satellite protein but we recently found that it localizes to the base region of the central pair of microtubules (which we term as CP-foot) with Centrin in multicilia (Zhao et al., 2021). Indeed, as we only confirmed α/β -tubulin mRNAs as ciliary mRNAs (Fig. 7, revised manuscript), other candidate mRNAs still require future investigations to clarify which ones are *bona fide* ciliary mRNAs.

As our reviewer #2 asks us to provide the information how many of the candidate mRNAs are already known to be located on the cilia, we further analyzed our deep sequencing results. Due to the update of gene ontology annotations in the AmiGo 2 database, the numbers of downregulated cilia-related genes upon FMRP depletion in Exp. 1 and Exp. 2 became 89 and 118 (in contrast to 91 and 117 in the original manuscript), respectively, and the overlapping genes became 36 (in contrast to 30 in the original manuscript) (Fig. 7c and Supplementary Data 1, revised manuscript). The majority of these candidate ciliary mRNAs (63/89 and 92/118) encoded proteins in ciliary shafts according to gene ontology annotations of the AmiGo 2 database (Supplementary Data 1). In addition to mRNAs of α - and β -tubulin isoforms, the 36 overlapped candidate mRNAs included those of MT regulators (*Dcdc2a* and *Dcx*) (Grati et al., 2015; Moores et al., 2004), tubulin polyglutamylase and glycyclase (*Till9* and *Till10*) (Ikegami and Setou, 2009; Konno et al., 2016), axonemal dynein subunits (*Dnah12* and *Dnali1*) (King, 2016), subunits of axonemal dynein-docking complexes (*Drc7* and *Ttc25*) (Morohoshi et al., 2020; Wallmeier et al., 2016), IFT subunits (*Bbs1*, *Ift27*, and *Ift52*) (Ishikawa and Marshall, 2017), other proteins involved in the structure and function of motile cilia (e.g., *Armc9*, *Cep131*, *Cfap69*, *Dnajb13*, and *Tmem17*) (Dong et al., 2018; El Khouri et al., 2016; Garcia-Gonzalo and Reiter, 2017; Latour et al., 2020; Zhao et al., 2021), and even FMRP (*Fmr1*) (Fig. 7c and Supplementary Table 1, revised manuscript).

We have modified the text to include the above information and discussed potential implications of these candidate mRNAs in the revised manuscript. We also point out that “future validation of *bona fide* ciliary mRNAs additional to those of tubulin and elucidation of

their translational regulations and physiological importance will lead to comprehensive understandings on roles of the ciliary local translation” in the Discussion.

Minor point:

It would be great to see supplemental material videos for the cilia degeneration experiments in the absence of FMRP.

Response:

Thanks for the suggestion. We have included the movie in the revised manuscript (Supplementary Movie 1).

It is unclear why different ribosomal proteins are analyzed in the western vs IF studies in Fig.1, some comment is required to help following the rationale.

Response:

For the immunoblotting results in Figure 1c, as total proteins from each preparation of purified cilia were only sufficient for one SDS-PAGE gel, we cut the blot horizontally into four segments according to positions of protein size markers. We chose to visualize RPL4, eIF3d, eIF3h, and Gapdh because their molecular masses allowed us to use each segment for one protein. The immunoblot for eIF3d was then stripped off antibodies and re-probed to visualize IFT81. Similar strategy was used to achieve immunoblotting results in Supplementary Figure 1a. We have clarified these in figure legends in the revised manuscript.

Antibodies used for IF studies were selected based on availability and compatibility with the IF procedure. Anti-RPL4 antibody, for instance, was good for immunoblotting but not for IF. To exclude staining artifact, we also confirmed ciliary localizations using GFP-tagged proteins (Fig. 1e). Due to the large number of components of the ribosome, we chose to verify one subunit for the large subunit (RPL11) and one for the small subunit (RPS3).

Figure 1A. Spelling error mass spectrometry

Response:

We are sorry for the typo. We have corrected it in the revised manuscript.

Page 3 Line 39. Protruding

Response:

We have corrected the typo.

Page 3 Line 54. Sperm uses flagella for locomotion not motile cilia

Response:

We have modified the sentence.

Page 3 Line 57.manifests as a syndromic disease

Response:

We have modified the sentence. Thanks.

References

- Buxbaum, A.R., Haimovich, G., and Singer, R.H. (2015). In the right place at the right time: visualizing and understanding mRNA localization. *Nat Rev Mol Cell Biol* *16*, 95-109.
- Darnell, J.C., Van Driesche, S.J., Zhang, C., Hung, K.Y., Mele, A., Fraser, C.E., Stone, E.F., Chen, C., Fak, J.J., Chi, S.W., *et al.* (2011). FMRP stalls ribosomal translocation on mRNAs linked to synaptic function and autism. *Cell* *146*, 247-261.
- Delgehyr, N., Meunier, A., Faucourt, M., Bosch Grau, M., Strehl, L., Janke, C., and Spassky, N. (2015). Ependymal cell differentiation, from monociliated to multiciliated cells. *Methods Cell Biol* *127*, 19-35.
- Dong, F.N., Amiri-Yekta, A., Martinez, G., Saut, A., Tek, J., Stouvenel, L., Lores, P., Karaouzene, T., Thierry-Mieg, N., Satre, V., *et al.* (2018). Absence of CFAP69 Causes Male Infertility due to Multiple Morphological Abnormalities of the Flagella in Human and Mouse. *Am J Hum Genet* *102*, 636-648.
- Duan, S., Li, H., Zhang, Y., Yang, S., Chen, Y., Qiu, B., Huang, C., Wang, J., Li, J., Zhu, X., *et al.* (2021). Rabl2 GTP hydrolysis licenses BBSome-mediated export to fine-tune ciliary signaling. *EMBO J* *40*, e105499.
- El Khouri, E., Thomas, L., Jeanson, L., Bequignon, E., Vallette, B., Duquesnoy, P., Montantin, G., Copin, B., Dastot-Le Moal, F., Blanchon, S., *et al.* (2016). Mutations in DNAJB13, Encoding an HSP40 Family Member, Cause Primary Ciliary Dyskinesia and Male Infertility. *Am J Hum Genet* *99*, 489-500.
- Enam, S.U., Zinshteyn, B., Goldman, D.H., Cassani, M., Livingston, N.M., Seydoux, G., and Green, R. (2020). Puromycin reactivity does not accurately localize translation at the subcellular level. *Elife* *9*.
- Garcia-Gonzalo, F.R., and Reiter, J.F. (2017). Open Sesame: How Transition Fibers and the Transition Zone Control Ciliary Composition. *Cold Spring Harb Perspect Biol* *9*.
- Goncalves, J., and Pelletier, L. (2017). The Ciliary Transition Zone: Finding the Pieces and Assembling the Gate. *Mol Cells* *40*, 243-253.
- Grati, M., Chakchouk, I., Ma, Q., Bensaid, M., Desmidt, A., Turki, N., Yan, D., Baanannou, A., Mittal, R., Driss, N., *et al.* (2015). A missense mutation in DCDC2 causes human recessive deafness DFNB66, likely by interfering with sensory hair cell and supporting cell cilia length regulation. *Hum Mol Genet* *24*, 2482-2491.
- Hafner, A.-S., Donlin-Asp, P.G., Leitch, B., Herzog, E., and Schuman, E.M. (2019). Local protein synthesis is a ubiquitous feature of neuronal pre- and postsynaptic compartments. *Science* *364*.
- Holt, C.E., Martin, K.C., and Schuman, E.M. (2019). Local translation in neurons: visualization and function. *Nat Struct Mol Biol* *26*, 557-566.
- Ikegami, K., and Setou, M. (2009). TTL10 can perform tubulin glycylation when co-expressed with TTL8. *Febs Lett* *583*, 1957-1963.
- Ishikawa, H., and Marshall, W.F. (2017). Intraflagellar Transport and Ciliary Dynamics. *Cold Spring Harb Perspect Biol* *9*.
- Ishikawa, T. (2017). Axoneme Structure from Motile Cilia. *Cold Spring Harb Perspect Biol* *9*.
- Jung, H., Yoon, B.C., and Holt, C.E. (2012). Axonal mRNA localization and local protein synthesis in nervous system assembly, maintenance and repair. *Nat Rev Neurosci* *13*, 308-324.
- King, S.M. (2016). Axonemal Dynein Arms. *Cold Spring Harb Perspect Biol* *8*.
- Konno, A., Ikegami, K., Konishi, Y., Yang, H.J., Abe, M., Yamazaki, M., Sakimura, K., Yao, I., Shiba, K., Inaba, K., *et al.* (2016). Ttl9(-/-) mice sperm flagella show shortening of doublet 7, reduction of doublet 5 polyglutamylolation and a stall in beating. *J Cell Sci* *129*, 2757-2766.

Laggerbauer, B., Ostareck, D., Keidel, E.M., Ostareck-Lederer, A., and Fischer, U. (2001). Evidence that fragile X mental retardation protein is a negative regulator of translation. *Hum Mol Genet* *10*, 329-338.

Latour, B.L., Van De Weghe, J.C., Rusterholz, T.D., Letteboer, S.J., Gomez, A., Shaheen, R., Gesemann, M., Karamzade, A., Asadollahi, M., Barroso-Gil, M., *et al.* (2020). Dysfunction of the ciliary ARMC9/TOGARAM1 protein module causes Joubert syndrome. *J Clin Invest* *130*, 4423-4439.

Moores, C.A., Perderiset, M., Francis, F., Chelly, J., Houdusse, A., and Milligan, R.A. (2004). Mechanism of microtubule stabilization by doublecortin. *Mol Cell* *14*, 833-839.

Morohoshi, A., Miyata, H., Shimada, K., Nozawa, K., Matsumura, T., Yanase, R., Shiba, K., Inaba, K., and Ikawa, M. (2020). Nexin-Dynein regulatory complex component DRC7 but not FBXL13 is required for sperm flagellum formation and male fertility in mice. *PLoS Genet* *16*, e1008585.

Mourao, A., Christensen, S.T., and Lorentzen, E. (2016). The intraflagellar transport machinery in ciliary signaling. *Curr Opin Struct Biol* *41*, 98-108.

Shigeoka, T., Jung, H., Jung, J., Turner-Bridger, B., Ohk, J., Lin, J.Q., Amieux, P.S., and Holt, C.E. (2016). Dynamic Axonal Translation in Developing and Mature Visual Circuits. *Cell* *166*, 181-192.

Wallmeier, J., Shiratori, H., Dougherty, G.W., Edelbusch, C., Hjeij, R., Loges, N.T., Menchen, T., Olbrich, H., Pennekamp, P., Raidt, J., *et al.* (2016). TTC25 Deficiency Results in Defects of the Outer Dynein Arm Docking Machinery and Primary Ciliary Dyskinesia with Left-Right Body Asymmetry Randomization. *Am J Hum Genet* *99*, 460-469.

Zhang, Y., Chen, Y., Zheng, J., Wang, J., Duan, S., Zhang, W., Yan, X., and Zhu, X. (2019). Vertebrate Dynein-f depends on Wdr78 for axonemal localization and is essential for ciliary beat. *J Mol Cell Biol* *11*, 383-394.

Zhao, H., Chen, Q., Li, F., Cui, L., Xie, L., Huang, Q., Liang, X., Zhou, J., Yan, X., and Zhu, X. (2021). Fibrogranular materials function as organizers to ensure the fidelity of multiciliary assembly. *Nat Commun* *12*, 1273.

Zheng, J., Liu, H., Zhu, L., Chen, Y., Zhao, H., Zhang, W., Li, F., Xie, L., Yan, X., and Zhu, X. (2019). Microtubule-bundling protein Spef1 enables mammalian ciliary central apparatus formation. *J Mol Cell Biol* *11*, 67-77.

Zhu, L., Liu, H., Chen, Y., Yan, X., and Zhu, X. (2019). Rsph9 is critical for ciliary radial spoke assembly and central pair microtubule stability. *Biol Cell* *111*, 29-38.

REVIEWERS' COMMENTS

Reviewer #1 (Remarks to the Author):

The authors have largely adequately addressed my concerns and I think the paper is acceptable. For the FMRP knock-down experiments, with the new information, a reduction in total mRNA FISH via, for example, polyA FISH, would be useful, but the sequencing experiments show that there is less mRNA after the FMRP knock-down.

Reviewer #2 (Remarks to the Author):

The revised manuscript resolved most of the issues that I have been concerned about. So, I think this can be accepted for publication, with one last request. Although the authors submitted the mass spec result as the source data, I still believe that the raw data must be shared as RNA-seq data through the GEO database. Therefore, I highly recommend that the authors submit their RAW files to the public repository, such as PRIDE or MassIVE DB.

Reviewer #4 (Remarks to the Author):

All concerns raised by the reviewer #3 have been successfully addressed. I think that the article has been nicely improved and warrant publication.

Point-to-point responses:

Reviewer #1:

The authors have largely adequately addressed my concerns and I think the paper is acceptable. For the FMRP knock-down experiments, with the new information, a reduction in total mRNA FISH via, for example, polyA FISH, would be useful, but the sequencing experiments show that there is less mRNA after the FMRP knock-down.

Response:

We thank our reviewer sincerely for helping us to improve the manuscript and appreciate the suggestion of polyA FISH. It has been shown that many dendritic mRNAs are deadenylated to become translationally incompetent and re-activated for local translation through cytoplasmic polyadenylation (Udagawa et al., 2012). Therefore, it is possible that many ciliary mRNAs also lack polyA tails and thus cannot be properly detected by polyA FISH. In this case, polyA FISH is unlikely suitable for assessing the changes in ciliary mRNA levels upon the depletion of *Fmrp*. As our reviewer has pointed out, our sequencing results already suggest a reduction in ciliary mRNAs upon the depletion of *Fmrp*.

Reviewer #2:

The revised manuscript resolved most of the issues that I have been concerned about. So, I think this can be accepted for publication, with one last request. Although the authors submitted the mass spec result as the source data, I still believe that the raw data must be shared as RNA-seq data through the GEO database. Therefore, I highly recommend that the authors submit their RAW files to the public repository, such as PRIDE or MassIVE DB.

Response:

We sincerely thank our reviewer for helping us to improve the manuscript. We have deposited the mass spectrometry results to public database as requested.

Reviewer #4:

All concerns raised by the reviewer #3 have been successfully addressed. I think that the article has been nicely improved and warrant publication.

Response:

We thank the reviewer for assessing our revised manuscript and appreciating our effects.

Reference:

Udagawa, T., Swanger, S.A., Takeuchi, K., Kim, J.H., Nalavadi, V., Shin, J., Lorenz, L.J., Zukin, R.S., Bassell, G.J., and Richter, J.D. (2012). Bidirectional control of mRNA translation and synaptic plasticity by the cytoplasmic polyadenylation complex. *Mol Cell* 47, 253-266.